# Chemical-induced phase transition and global conformational reorganization of chromatin

Tengfei Wang[1,9], Shuxiang Shi[1,2,9], Yuanyuan Shi[3,4,9], Peipei Jiang[3,4,9], Ganlu Hu[5,9], Qinying Ye [1], Zhan Shi[6], Kexin Yu[1,7], Chenguang Wang[3,4,8], Guoping Fan [5], Suwen Zhao [1,7], Hanhui Ma [1], Alex C. Y. Chang [3,4,8], Zhi Li [6], Qian Bian [3,4] ✉ & Chao-Po Lin [1] ✉

Chemicals or drugs can accumulate within biomolecular condensates formed through phase separation in cells. Here, we use super-resolution imaging to search for chemicals that induce phase transition within chromatin at the microscale. This microscopic screening approach reveals that adriamycin (doxorubicin) − a widely used anticancer drug that is known to interact with chromatin − specifically induces visible local condensation and global conformational change of chromatin in cancer and primary cells. Hi-C and ATAC-seq experiments systematically and quantitatively demonstrate that adriamycin-induced chromatin condensation is accompanied by weakened chromatin interaction within topologically associated domains, compartment A/B switching, lower chromatin accessibility, and corresponding transcriptomic changes. Mechanistically, adriamycin complexes with histone H1 and induces phase transition of H1, forming fibrous aggregates in vitro. These results reveal a phase separation-driven mechanism for a chemotherapeutic drug.

Phase separation refers to the de-mixing process by which components in a mixed solution segregate into two or more phases with their own uniform properties. Interactions between hydrogen bonds, cations, charges, pi, or van der Waals between solute molecules could all drive phase separation processes[1]. In a simple form called liquid–liquid phase separation, a solute can be de-mixed from the solvent, forming a dense phase of condensates. Liquid–liquid phase separation of biomolecules, mostly proteins and/or nucleic acids, has been employed to explain the formation, maintenance, and function of membrane-less organelles, such as germ P granules[2], stress granules and processing bodies (P-bodies)[3,4], Cajal bodies[5], PML bodies[6], paraspeckles[7], and nucleoli[8]. Recently, biomolecule condensates have also been demonstrated to involve in more diverse processes, such as DNA repair[9], T cell activation[10], signal transduction[11], neurodegenerative diseases[12], and osmotic pressure response[13].

The physical properties of biomolecule condensates are determined by both intrinsic properties of biomolecules and environmental factors, such as pH, salt concentrations, or modifications on biomolecules[14]. For example, FUS forms liquid-like droplets at the physiological concentration but converts to solid-like fibrous

[1]School of Life Science and Technology, ShanghaiTech University, 201210 Shanghai, China. [2]Lingang Laboratory, 200031 Shanghai, China. [3]Shanghai Institute of Precision Medicine, Ninth People's Hospital, Shanghai Jiao Tong University School of Medicine, 200125 Shanghai, China. [4]Shanghai Key Laboratory of Reproductive Medicine, Shanghai Jiao Tong University School of Medicine, 200025 Shanghai, China. [5]Shanghai Institute for Advanced Immunochemical Studies, ShanghaiTech University, 201210 Shanghai, China. [6]School of Physical Science and Technology, ShanghaiTech University, 201210 Shanghai, China. [7]iHuman Institute, ShanghaiTech University, 201010 Shanghai, China. [8]Department of Cardiology, Ninth People's Hospital, Shanghai Jiao Tong University School of Medicine, Shanghai, China. [9]These authors contributed equally: Tengfei Wang, Shuxiang Shi, Yuanyuan Shi, Peipei Jiang, Ganlu Hu. ✉e-mail: qianbian@shsmu.edu.cn; linzhb@shanghaitech.edu.cn

aggregates with disease-associated mutations[15]. Another example is the debate on the (liquid or solid) state of chromatin[16]. Multiple lines of evidence suggest that chromatin exhibits a liquid-like state, given the observation that purified chromatin undergoes liquid–liquid phase separation in microinjected cells, producing dynamic droplets[17]. The formation of droplets in vitro and in vivo is dependent on linker sizes of DNA, the linker histone H1, the acetylation of histones, and the presence of chromatin-associated proteins such as BRD4[17]. Consistently, as revealed by super-resolution microscopy, chromatin domains exhibited fast motion dynamics, while heterochromatin-rich regions showed slow dynamics in living cells[17,18]. Importantly, CBX2 or heterochromatin protein HP1alpha (CBX5) formed liquid condensates in vitro and have been proposed to drive the formation of heterochromatin[19–21]. Together, those results support the liquid-like property of euchromatin and heterochromatin.

On the other hand, the chromatin could also adapt to different physical properties. In vitro, the chromatin can form aggregates with either liquid, gel-like, or solid properties depending on the salt concentration, nucleosome composition, and histone modifications. Strickfaden et al. proposed that the heterochromatin was physically constrained, forming solid-like or gel-like scaffolds at the mesoscale (10–1000 nm)[22]. Moreover, the gel-like nature of chromatin was not altered by intracellular osmolarity, histone acetylation, or inhibition of bromodomain proteins that mediate phase separation[22]. It's possible that both liquid and solid reflect the viscoelastic (rheological) nature of the chromatin fiber[23]. The chromatin polymer, therefore, can be "elastic solid" or "viscous liquid," depending on different timescales, length scales, and biological processes such as transcription, replication, or DNA repair[23]. Nonetheless, how the physical property of chromatin influences the three-dimensional conformation, gene expression, or biological effect remains to be investigated.

In the present study, we employed a highly specific nucleic acid dye and super-resolution microscopy to screen for small chemicals that can interrogate the physical state of the viscoelastic chromatin. This phenotypical screening reveals an anthracycline antibiotic and chemotherapeutic drug known to bind and interfere with the chromatin, adriamycin (doxorubicin), induces the condensation of chromatin in live and fixed cells in a visible, reversible, and cell-type-specific manner. This condensation effect of adriamycin on the chromatin is independent of its topoisomerase II poisoning activity, DNA damage, reactive oxygen species, and programmed cell death pathways. By performing Hi-C and ATAC-Seq, we demonstrated that adriamycin induces global changes on chromatin conformation, including the loss of topological associating domain (TAD) boundaries, alterations of chromatin accessibilities, and shifted expression patterns of coding genes and transposable elements. Mechanistically, adriamycin complexes with and induces phase transition of the linker histone H1, resulting in the condensation of native chromatin. Together, our results reveal a phase separation-driven mechanism of a chemotherapeutic drug.

## Results

### Adriamycin induces chromatin condensation in cancer and primary cells

In an attempt to manipulate the physical property of chromatin in cells, we employed stimulated emission depletion (STED) microscopy[24] to examine changes in chromatin structures upon treatment with small chemicals using the DNA-specific, nontoxic far-red DNA dye, SiR-Hoechst (SiR), for DNA nanoscopy[25]. Chromatin within U2OS osteoblastoma cells was fixed and examined after short-term treatment with a series of chemicals or chemotherapeutic drugs targeting genomic DNA, including the topoisomerase I poison camptothecin[26], topoisomerase II poisons etoposide[27] and adriamycin[28], the radiomimetic DNA cleaving agent bleomycin[29], as well as DNA intercalators/adduct-inducers psoralen, cisplatin, and cytophosphane (cyclophosphamide)[30–32]. We observed condensed chromatin puncta only upon adriamycin

treatment in U2OS, HCT116 and HeLa cells (Fig. 1a, b). To examine whether this is a common phenomenon, we synchronized U2OS cells by RO-3306, a G2/M phase inhibitor, followed by releasing in RO-3306-free medium, which resulted in cell cycle progression to different stages (Supplementary Fig. 1a). In undisturbed U2OS, ~80% of cells exhibited the punctate pattern of genomic DNA upon adriamycin treatment, correlated with the percentage of cells in the G1/S phase. Stalling at the G2/M phase (4 h after releasing) decreased the percentage of cells with condensed chromatin puncta to ~50%, which was gradually increased again with further cell cycle progression (i.e., the decrease of G2/M percentage) (Supplementary Fig. 1b, c). Thus, almost all cells in the G1/S phase were responsive to adriamycin treatment and showed the punctate pattern of genomic DNA.

We next examined whether chromatin condensates induced by adriamycin were the consequence of programmed cell death. Short-term (4 h) treatment of adriamycin at 1.5 μg/ml, a concentration close to peak plasma levels (1.64 ± 0.47 μg/ml) after chemotherapy administration[33], did not induce apoptosis compared with sorafenib or TNFα/cycloheximide treatment for 16 h (Supplementary Fig. 2a). Importantly, apoptotic U2OS cells did not exhibit the punctate DNA pattern as adriamycin (Supplementary Fig. 2b, c). Moreover, none of the apoptosis inhibitor z-VAD-fmk, necrosis inhibitor necrostatin-1 (NEC-1), and ferroptosis inhibitor ferrostatin-1 (FER-1) blocked the formation of the punctate pattern of genomic DNA (Supplementary Fig. 2c). Therefore, the punctate DNA induced by adriamycin is less likely the consequence of programmed cell death.

It has been proposed that the pharmacological or cardiotoxic effects of adriamycin are the consequences of elevated reactive oxygen species (ROS) or ferroptosis[34,35]. Yet, neither the ROS producer hydrogen peroxide nor the ferroptosis inducer erastin[36] induced similar morphological change of DNA in cells (Fig. 1a), suggesting that adriamycin-induced formation of chromatin puncta is independent of ROS or ferroptosis.

We further characterized the condensed puncta in adriamycin-treated U2OS cells (Fig. 1c, d). The diameters of chromatin punctate induced by adriamycin ranged from 0.5 to 4 μm, ~10-fold larger compared to chromatin "dots" in control samples which were 0.08–0.2 μm in diameter as revealed by software measurement of Fig. 1c. To quantify chromatin distribution in control and adriamycin-treated samples, we calculated the radial distribution function, or RDF (the density of chromatin dots in a circular ring), as well as L-function (the measurement of the size of the cluster and the degree of condensation)[18,37] under each treatment condition in Fig. 1a (Supplementary Fig. 3a). The L-function shows a single peak at ~400 nm for adriamycin-treated cells, indicating the average size of condensed clusters (Fig. 1d). Analysis of the RDF and L-function further confirmed that adriamycin is the only drug that can induce chromatin condensation among those tested here (Supplementary Fig. 3b, c).

To examine the detailed structures of chromatin condensates induced by adriamycin, we performed electron microscopy (EM) on interphase U2OS nuclei. Under control conditions, interphase chromatin was relatively homogenous, mostly dispersal, and existed in fibrous structures throughout the nucleoplasm (Fig. 1e). Treatment with adriamycin induced heterogeneous, irregular, dense chromatin structures in U2OS nucleoplasm with 0.1–2.6 μm in width (Fig. 1e). Smaller, lower-density chromatin structures exhibiting fibrous morphology could also be observed. Adriamycin also induced larger, high-density structures located in close proximity to the nuclear lamina where the heterochromatin is associated (Fig. 1f). Those results suggest that the dense chromatin structures induced by adriamycin are related to heterochromatin.

Adriamycin exhibits potent therapeutic effects against cancer cells and clinical cardiotoxicity against cardiomyocytes[38,39]. To test the cell-type-specific effects of adriamycin-induced alteration of chromatin structures, we isolated primary cardiomyocytes and hepatocytes

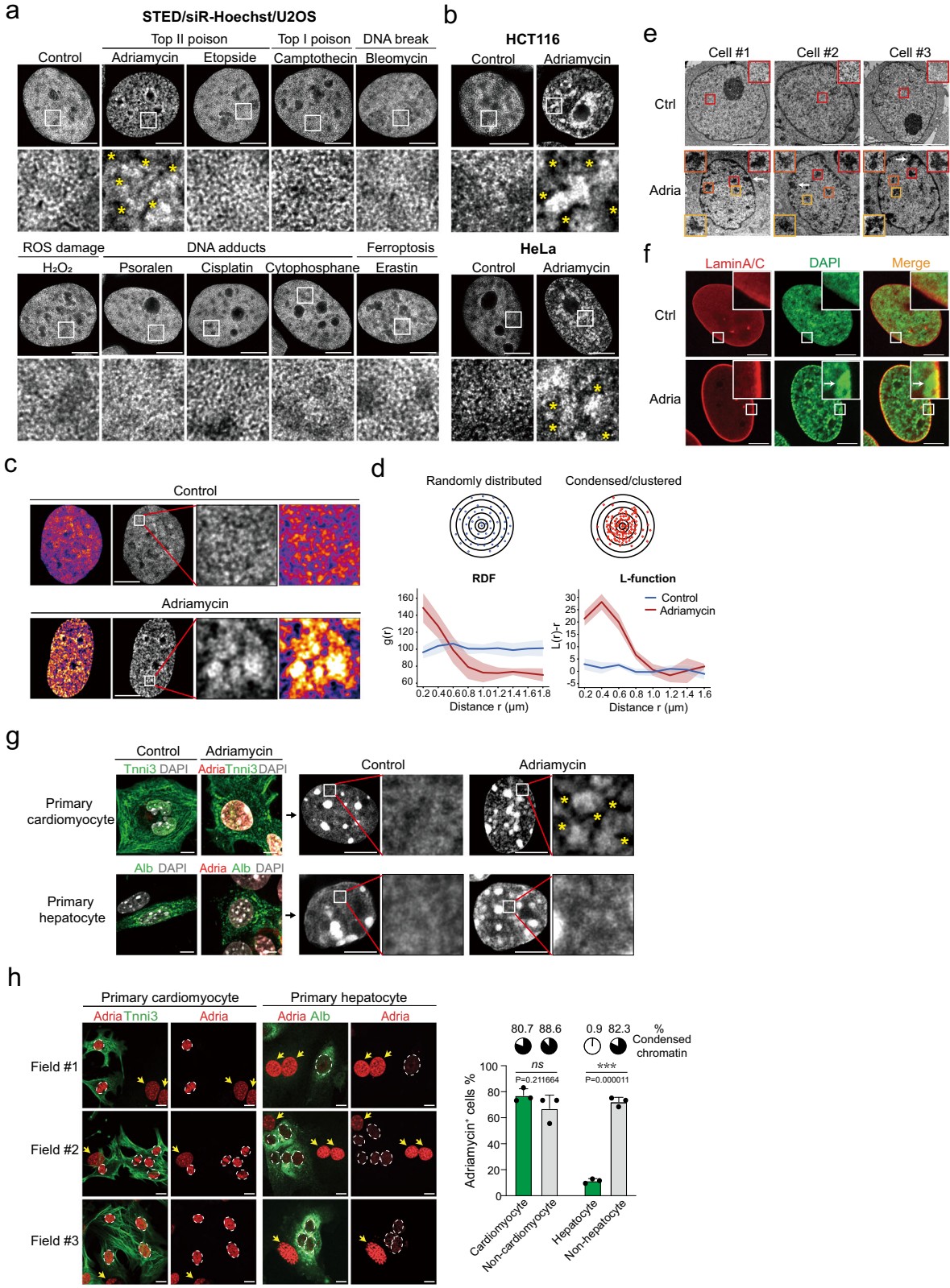

from P1 neonatal mice. Isolated cardiomyocytes and hepatocytes expressed the specific markers, Troponin I (Tnni3) and Albumin (Alb), respectively (Fig. 1g). Adriamycin induced chromatin morphological alteration, forming dense structures in cardiomyocytes but not in hepatocytes (Fig. 1g). Adriamycin also accumulated at significantly lower level in hepatocytes compared with adjacent non-hepatocytes (Fig. 1h), possibly due to the effluxion ability of hepatocytes[40,41], while

both cardiomyocytes and non-cardiomyocytes exhibited similar intranuclear level of adriamycin (Fig. 1h). To further investigate the correlation between chromatin condensation and cardiotoxicity, we treated primary cardiomyocytes with two clinically-used anthracyclines with high and low cardiotoxicity, daunorubicin and aclarubicin, respectively[42] (Supplementary Fig. 3d). Daunorubicin, similarly to adriamycin, induced chromatin condensation in primary

**Fig. 1 | Adriamycin induces chromatin condensation in cancer and primary cells. a**, **b** Screening of chemicals that alter chromatin structures at mesoscale by super-resolution microscopy. Only adriamycin, but not the other TOP2 poison etoposide, resulted in a punctate pattern of chromatin in U2OS cells (**a**). Similar phenomenon could also be observed in HCT116 and HeLa cells (**b**). Scale bar, 10 μm. Asterisk, low-density DNA regions. **c** Heatmaps of STED images demonstrated the differential distribution of DNA in control (randomly distributed) and adriamycin-treated samples. Scale bar, 10 μm. **d** Quantification of the degree of clustering by RDF and L-function (see "Methods" and Supplementary Fig. 3). Adriamycin induced significant clustering of chromatin compared to the control. The ribbon plots are employed to show means +/− SD (the width of ribbons; $n = 10$ cells). **e** Electron microscopy revealed different morphologies of chromatin condensates. Scale bar, 10 μm. White arrows indicate the condensates formed in the proximity of nuclear envelops. **f** Immunostaining of Lamin A/C (red) with DAPI staining (green) in U2OS cells. Scale bar, 10 μm. White arrows indicate the condensates formed in the proximity of nuclear envelops. Ctrl control, adria adriamycin. **g**, **h** Adriamycin induced significant chromatin condensation in primary cardiomyocytes, but to a much less extent in primary hepatocytes. **g** Primary cells isolated from P1 mice were cultured for 3–5 days, followed by 1.5 μg/ml adriamycin treatment for 4 h. Cells were immunostained with cardiomyocyte (Tnni3) and hepatocyte (Alb) markers, as well as DAPI. Cells positive for adriamycin were examined for their chromatin conformation. **h** The differential accumulation of adriamycin in different cell populations. Dashed circles indicate the nuclear outlines of cardiomyocytes (Tnni3$^+$) or hepatocytes (Alb$^+$). Yellow arrows indicate non-cardiomyocytes (Tnni3$^−$) or non-hepatocytes (Alb$^−$). Scale bar, 10 μm. For each group in the bar graph, three fields ($n = 3$) were calculated and totally 78, 79, 109, and 79 cells were counted for each group. Data are presented as means +/− SD in the bar graph. The ratios of cells showing condensed chromatin (as in Fig. 1g) in those four populations are shown above each bar. Ns not significant; ***$P < 0.001$ (two-tailed unpaired $t$ test). Source data are provided as a Source data file. Experiments of (**a**–**c**, **g**, **h**) were repeated at least three times, and experiments of (**e**) and (**f**) were repeated twice with similar results.

cardiomyocytes, while the effect of aclarubicin was limited (Supplementary Fig. 3d), suggesting a potential link between chromatin condensation and cardiotoxicity.

## Adriamycin co-localizes with chromatin condensates

The presence and dynamic of adriamycin can be traced by microscopy with its unique nature of UV absorption wavelength (~430 nm) and red fluorescence emission (~514 nm). We, therefore, aimed to follow the real-time dynamic and localization of adriamycin using Airyscan2 super-resolution microscopy (~120 nm resolution)[43]. In mouse embryonic fibroblasts (MEFs), adriamycin appeared as fibrous structures in nuclei within 30 min, then formed larger, condensed aggregates (described as adriamycin condensates hereafter) within 60 min (Supplementary Movie 1). The morphological transition of adriamycin condensates was more evident using STED microscopy (100 nm resolution) on cells fixed at different time points (Fig. 2a). For comparison, we labeled cisplatin—which also intercalates with DNA—with Texas Red[1] (Supplementary Fig. 4). In contrast to adriamycin, Texas Red-cisplatin accumulated with a more homogenous pattern in the nucleoplasm within a 60 min period (Fig. 2a). The fluidity of adriamycin within those condensates was further analyzed using fluorescence recovery after photobleaching (FRAP). A continuous increase in fluorescence recovery level over a 60 s period suggested that adriamycin within condensates could diffuse and exchange with adriamycin molecules present in the nucleoplasm (Fig. 2b and Supplementary Movie 2). Together, these results document the dynamics of adriamycin condensates in cells and indicate the reversibility of adriamycin association with condensates.

We next examined whether DNA condensates and adriamycin condensates were co-localized. In both U2OS and HCT116 cells, we observed strong overlapping between chromatin and adriamycin condensates 4 h after treatment (Fig. 2c). Importantly, adriamycin–DNA condensates could also be observed in live U2OS cells by the structured illumination microscopy (SIM) super-resolution microscopy (Fig. 2d). To further determine the dynamics and the property of adriamycin-chromatin condensates, we removed adriamycin from the culture medium after 4-h treatment. Surprisingly, the depletion of adriamycin nearly completely reversed the dense, punctate morphology of chromatin back to the homogenous pattern like the control (Fig. 2e). Therefore, despite of being shown to be gel/solid-like, the chromatin DNA is still capable of undergoing global re-localization at the microscale. This result, again, excludes the possibility that chromatin condensates were the secondary effect of programmed cell death.

## Adriamycin–chromatin condensates are irrelevant to DNA damage or transcription inhibition

Small chemicals, such as cisplatin or tamoxifen, could be partitioned within phase-separated condensates formed by the Mediator of RNA polymerase II transcription subunit I (MED1)[44], which associate with super enhancers[45]. We therefore asked if adriamycin co-localizes with MED1. We found that adriamycin condensates did not co-colocalize with MED1 foci (Supplementary Fig. 5a). DNA damage response (DDR) proteins have also been reported to form phase-separated condensates[9,46]. We excluded the possibility that adriamycin condensates are partitioned within those DDR condensates with the following evidence: first, adriamycin did not co-localize with double-strand break foci marked by γ-H2AX (Supplementary Fig. 5b). Second, at the concentrations we used, adriamycin induced less γ-H2AX than camptothecin or etoposide (Supplementary Fig. 5c) in both U2OS cells and human mesenchymal stem cells (MSCs), whereas neither camptothecin nor etoposide induced chromatin condensation in those two cell types (Supplementary Fig. 5d). Lastly, 53BP1, a DDR protein that has been demonstrated to form phase separation-mediated condensates, did not co-localize with adriamycin condensates (Supplementary Fig. 5b).

Inhibition of RNA polymerase II can lead to chromatin condensation or the formation of condensates comprised of SFPQ, NONO, FUS, and TAF15[47,48]. Although adriamycin can also inhibit transcription due to the formation of topoisomerase II cleavable complexes[49], treatment with the transcriptional inhibitors 5,6-dichloro-1-β-D-ribofuranosylbenzimidazole (DRB)[47] or THZ1[50] did not result in chromatin condensation at the microscale (Supplementary Fig. 5e). Together, those results suggest that adriamycin-induced chromatin condensation is independent of the condensates formed by the transcription machineries or by transcriptional pausing.

## Adriamycin induces chromatin condensation in a TOP2 isozyme-independent manner

Adriamycin forms a ternary complex with TOP2-DNA and inhibits TOP2's DNA re-ligation activity[51]. The subsequent degradation of TOP2 complexes exposes DNA ends, eliciting the DNA damage responses[52] (Supplementary Fig. 6a). To see if the adriamycin-induced chromatin condensates are dependent on the formation or degradation of TOP2 cleavable complexes, we pre-treated U2OS cells with dexrazoxane (ICRF-187), which prevents the formation of TOP2 cleavable complexes or the degradation of TOP2[53] (Supplementary Fig. 6a). Treatment with dexrazoxane did not block the formation of adriamycin-induced chromatin condensates, similar to the effect of pretreatment with proteasome inhibitor MG132 (Supplementary Fig. 6b). In addition, knockdown of TOP2β, the major isozyme subjected to adriamycin-mediated degradation in the interphase, did not block the formation of chromatin condensates (Supplementary Fig. 6c). Together, those results suggest that the TOP2-poisoning activity of adriamycin is not sufficient to drive chromatin condensation, consistent with the observation that the more specific TOP2 poison etoposide did not induce condensate formation (Fig. 1a).

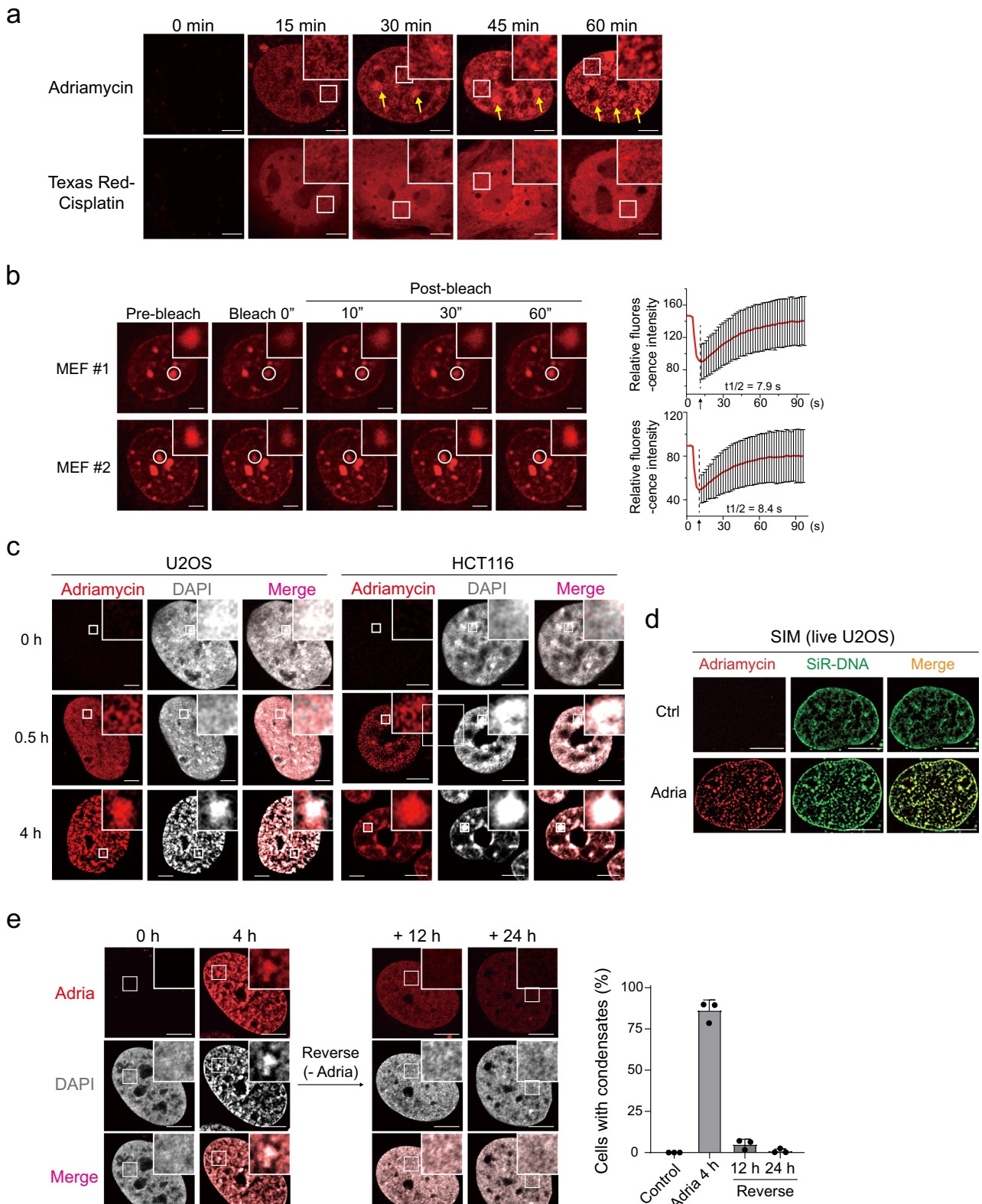

## Adriamycin predominantly accumulates within heterochromatin

We next examined the composition of chromatin within the condensates induced by adriamycin. To distinguish heterochromatin and euchromatin in living cells, we pulse-labeled chromatin with dUTP-Cy5[22,54]. With this approach, heterochromatin would be labeled as bright, large puncta if dUTP-Cy5 was incorporated into DNA during the mid/late S phase (Fig. 3a). Euchromatin would be labeled if dUTP-Cy5 was incorporated into DNA during the early S phase, appearing as hundreds of smaller foci scattered throughout the nucleoplasm (Fig. 3b). We observed clear co-localization between dUTP-Cy5-labeled heterochromatin and adriamycin (Fig. 3a, arrows), while few overlapping signals were observed between the euchromatin and adriamycin (Fig. 3b), suggesting that heterochromatin could be the primary

**Fig. 2 | Adriamycin forms complexes with chromatin condensates.**
**a** Comparison of dynamics of two DNA-intercalating reagents, adriamycin and cisplatin, in mouse embryonic fibroblasts (MEFs). As revealed by Airyscan2 microscope, adriamycin entered nuclei exhibited the fibrous morphology, followed by a transition towards dense aggregates. Cisplatin was conjugated with Texas Red for visualization while adriamycin has intrinsic excitation wavelength at 514 nm. Arrows, large puncta which could possibly be heterochromatin. Scale bar, 10 μm. **b** FRAP analysis showed the material exchange of adriamycin within condensates. Scale bar, 10 μm. Signals were corrected for photobleaching using a similarly sized unbleached area and then normalized to the ratio between the average intensity of the pre-bleach images and the lowest post-bleach intensity. The signal intensities are presented as means +/− standard deviation (SD) ($n = 10$–15 cells per condition). **c** Adriamycin co-localized with condensed chromatin in U2OS and HCT116 cells.

Adriamycin appeared fibrous or dotted at the 0.5 h time point, while the chromatin appeared homogenous. The co-localization became apparent at the 4 h time point. Scale bar, 10 μm. **d** Live imaging of the adriamycin–DNA condensates by the SIM super-resolution microscope. U2OS cells were incubated with 0.5 μM SiR-Hoechst overnight and treated with 1.5 μg/ml for 2 h before imaging. Scale bar, 10 μm. **e** Co-aggregated adriamycin and chromatin were reversible. U2OS cells were treated with 1.5 μg/ml adriamycin for 4 h, followed by drug removal for 12 and 24 h. The punctate pattern of adriamycin–DNA was almost completely disappeared after 24 h reversal. Scale bar, 10 μm. Quantification of % of punctate cells in each condition was performed on 3 separate fields ($n = 3$), each containing ~50 cells. Data are presented as means +/− SD. Ctrl, control; adria, adriamycin. Source data are provided as a Source data file. Experiments of (**a**) and (**b**) were repeated at least three times, and experiments of (**c**) and (**e**) were repeated twice with similar results.

location for chromatin condensates to form. However, it is worth noting that due to the photobleaching effect of microscopic live imaging, the association between adriamycin and the euchromatin could be underestimated in this analysis. We further examined the relative distribution of adriamycin condensates and heterochromatin and euchromatin markers. Heterochromatin protein 1 alpha (HP1a/ CBX5) and the heterochromatin marker H3K9me3 also co-localized with adriamycin condensates (Fig. 3c, d), consistent with the live imaging results. Interestingly, for euchromatin markers, adriamycin condensates exhibited a mutually exclusive pattern with H3K4me3 (Fig. 3e, promoters) but a partially overlapped pattern with H3K27ac (Fig. 3f, promoters and enhancers). Moreover, although the majority of adriamycin distributed in a pattern not overlapped with nuclear p53, some p53 molecules still located within adriamycin condensates (Fig. 3g, arrows). Together, these results suggest that adriamycin predominantly associates with heterochromatin in cells, despite of evident euchromatin association.

## Adriamycin induces global changes in chromatin accessibility
Having established that adriamycin induced significant chromatin structure reorganization at the microscopic level, we next sought to understand the nature of these changes at the molecular level, and to explore the functional consequences of these chromatin structural changes, particularly in transcriptional regulation. To this end, we performed ATAC-Seq (assay for transposase-accessible chromatin with high-throughput sequencing) to examine the open chromatin landscape in etoposide- or adriamycin-treated cells, both of which are TOP2 poisons while only the latter induces the formation of chromatin condensates. At the concentration we used to treat U2OS cells, etoposide resulted in 496 up- and 853 down-regulated genes at the transcriptional level, but only 150 gained and 72 lost peaks (Fig. 4a). In contrast, adriamycin treatment resulted in a greater alterations in transcription (414 up-regulated and 2096 down-regulated genes) and a lot more shifts in chromatin accessibility (9456 gained and 11,747 lost peaks), suggesting an extensive modulation of chromatin accessibility (Fig. 4a). Most of gained/lost peaks are located at promoters, as well as introns and distal intergenic regions where enhancers are located (Supplementary Fig. 7a). Heatmaps of peaks near transcriptional start sites (TSSs) revealed decreased peak heights on accessible sites, suggesting a general loss of chromatin accessibility at TSSs (Fig. 4b). We also analyzed all peaks located across the whole genome, and categorized them into gained, lost, and unchanged groups (Supplementary Fig. 7b). For the whole genomic regions, adriamycin-treatment resulted in both gain and loss of peaks, while the extent of "gain" is obviously less than that of "loss" (Supplementary Fig. 7b). Together, those results indicate that adriamycin-treatment resulted in global changes in chromatin accessibility with a general trend of decreased chromatin accessibility, especially at TSS sites (Supplementary Fig. 7c). Given that the ATAC-Seq peaks are predominantly located within euchromatic regions, these data suggest that adriamycin-treatment profoundly influence the organization of euchromatin.

We further analyzed the transcriptional profiles the pathway enrichment in etoposide- or adriamycin-treated U2OS cells. Both etoposide and adriamycin activated biological pathways related to DNA damage stress, including E2F targets, DNA repair, and apoptosis (Supplementary Fig. 8a and Supplementary Data 1). Interestingly, although both etoposide and adriamycin treatment share p53-activated or p53-repressed targets, some p53 pathway genes are more strongly activated by etoposide (Supplementary Fig. 8b), consistent with the stronger γ-H2AX activation by etoposide compared with adriamycin (Supplementary Fig. 5c). Correlation analysis of RNA-Seq and ATAC-Seq revealed that genes specifically activated by etoposide, exemplified by p53-regulated tumor suppressor BTG2, are independent of changes in ATAC-seq peak signals (Supplementary Fig. 8c, d). In contrast, increased accessibility at CTRB1 promoter region upon adriamycin treatment is correlated with aberrant activation of CTRB1 (Supplementary Fig. 8d), suggesting that adriamycin-induced increase of chromatin accessibility could cause the activation of specific genes, albeit to a lesser extent. Importantly, a large portion of genes downregulated by adriamycin treatment was correlated with the loss of chromatin accessibility at the promoter region (Supplementary Fig. 8c), as exemplified by TNFAIP1 and FOS (Fig. 4c), which affect cell growth, survival, and transformation[55,56]. Hence, significant biological consequences could result from the adriamycin-induced loss of chromatin accessibility.

## Adriamycin induces global changes in higher-order chromatin organization
We next performed Hi-C to evaluate the impact of adriamycin on higher-order chromatin organization. Hi-C heatmaps revealed pronounced changes in chromatin interaction patterns between control and adriamycin-treated U2OS cells. Upon adriamycin treatment, the near-diagonal signals appeared less prominent on Hi-C heatmaps (Fig. 5a), indicating the loss of chromatin compaction at shorter ranges. We quantified the chromatin contact probability on all chromosomes as a function of genomic separation (P(s)). Comparison between the P(s) curves in control and adriamycin-treated cells showed that the chromatin contact probability between 50 and 500 kb range is decreased upon adriamycin treatment (Fig. 5b). In contrast, chromatin contact probability between 1 and 10 Mb exhibits a moderate increase (Fig. 5b). Thus, adriamycin treatment led to scale-dependent changes in the overall chromatin compaction.

Interphase chromatin folds into TADs of sub-megabase sizes, which appear as dense squares along the diagonals of Hi-C interaction heatmaps. Notably, the TAD structure became significantly perturbed upon adriamycin treatment, with the TAD boundaries diminishing and the neighboring TAD frequently merging (Fig. 5a). We further quantified the changes in the TAD organization using an insulation-index-based approach (Fig. 5c). For any given 10 kb genomic bin, the insulation score was calculated by aggregating the interactions occurring across the bin within a 500 kb window. Thus, lower insulation scores of genomic regions indicate greater insulation abilities, with the local

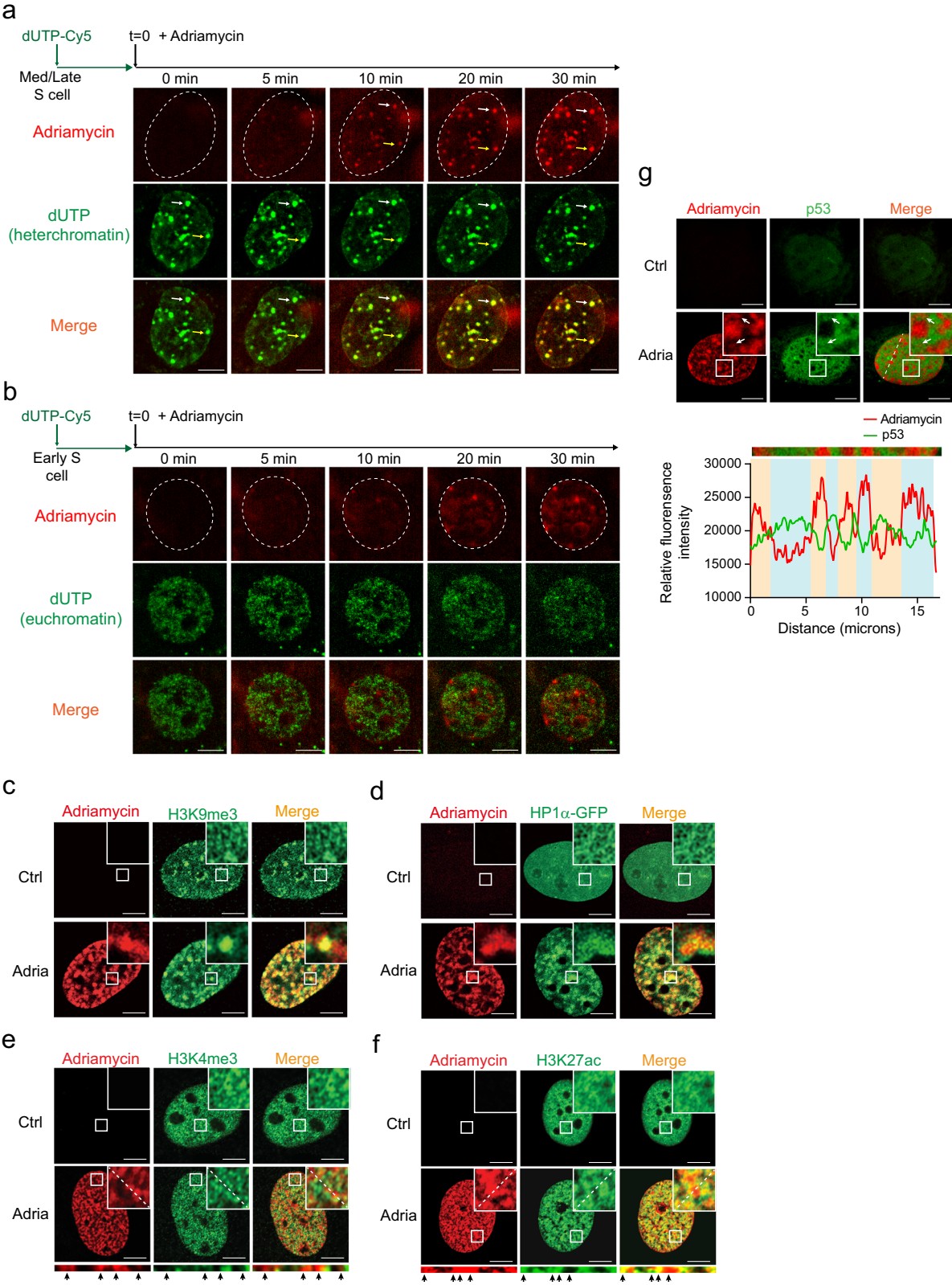

minima of the insulation profile denoting TAD boundaries. Using this approach, we identified 3179 and 2571 TAD boundaries in control and adriamycin-treated U2OS cells, respectively (Fig. 5d). The TAD boundaries in adriamycin-treated cells exhibited higher insulation scores and lower boundary strength scores compared to those in control cells. Among the TAD boundaries, 1029 were specifically identified in control cells but lost upon adriamycin treatment, 2150

were identified in both samples, and 421 were specifically identified in adriamycin-treated cells (Fig. 5d). We generated the averaged insulation profiles around the three groups of TAD boundaries to further assess their changes upon adriamycin treatment. Both the control-specific and shared TAD boundaries exhibited significant increases in insulation scores, indicating diminished insulation ability (Supplementary Fig. 9a). In contrast, the adriamycin-specific boundaries only

**Fig. 3 | Adriamycin–chromatin condensates co-localize with heterochromatin.**
**a**, **b** U2OS cells were pulse-labeled with dUTP-Cy5 to label heterochromatin (**a**) or euchromatin (**b**), depending on the status of cell cycle. Cells were live-imaged at ×63 upon 1.5 µg/ml adriamycin treatment on Leica Thunder Imager microscope. As heterochromatin regions indicated by arrows, adriamycin accumulated primarily in heterochromatin, not euchromatin regions. Scale bar, 10 µm.
**c**, **d** Adriamycin–chromatin condensates were co-localized with heterochromatin marker H3K9me3 (**c**) and transfected HP1-CFP (**d**). Scale bar, 10 µm. **e**, **f** Adriamycin condensates were partially co-localized with euchromatin marker H3K27ac (promoters and enhancers), but were localized mutual exclusively with H3K4me3 (promoters). Arrows indicate adriamycin condensates. Scale bar, 10 µm.
**g** Adriamycin-induced nuclear p53 and adriamycin mainly exhibited mutually exclusive occupancy in nuclei. Arrows indicate the minority p53 that located within adriamycin condensates. Scale bar, 10 µm. Source data are provided as a Source data file. All experiments were repeated twice with similar results.

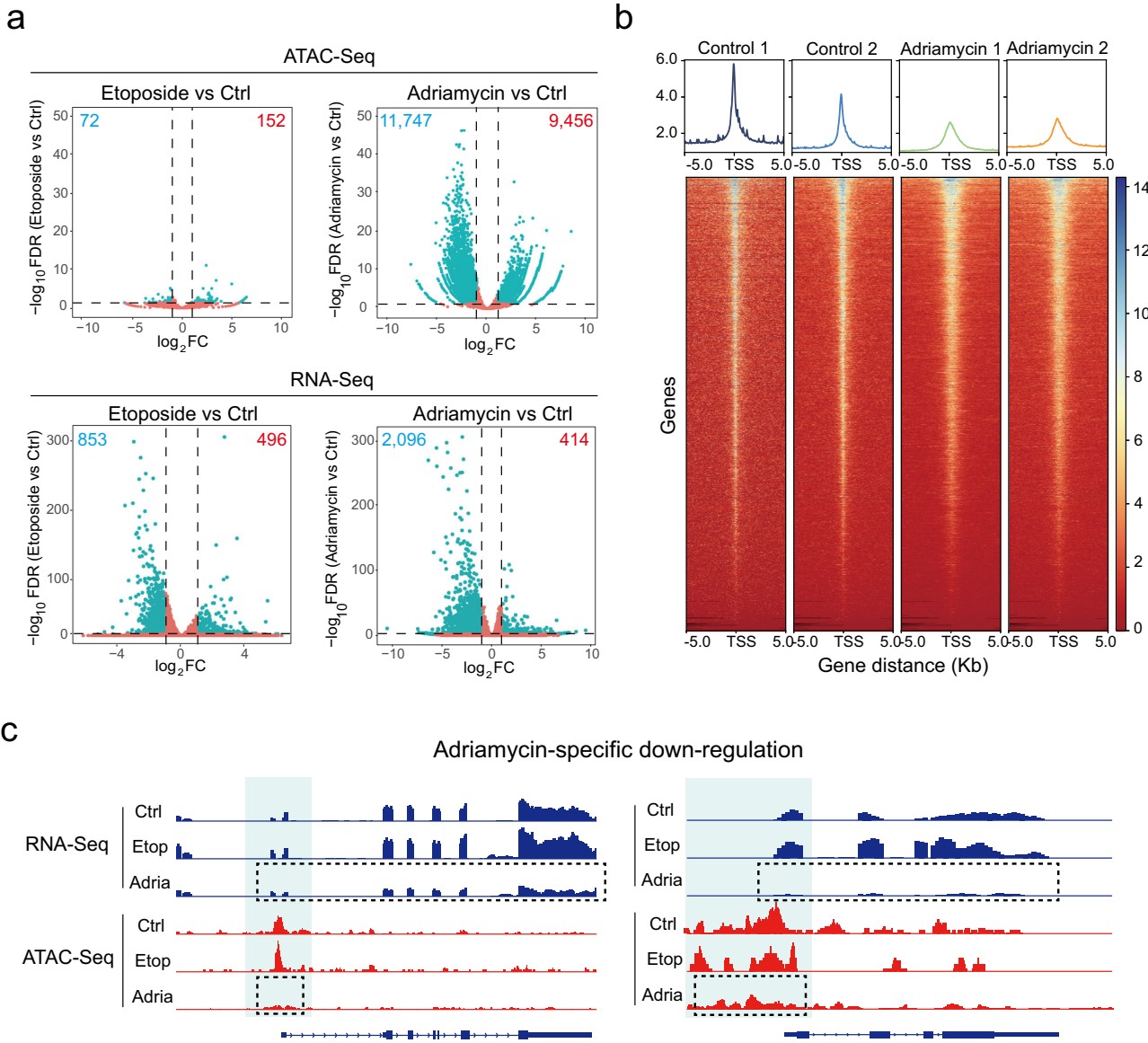

**Fig. 4 | Adriamycin induces genome-wide changes in chromatin accessibility.**
**a** Volcano plots showed the changes of ATAC-Seq peaks (top row) or gene expression (bottom row) upon etoposide (left) or adriamycin (right) treatment. Adriamycin, but not etoposide, induced a large amount of gain/loss peaks.
**b** Averaged line graph and heatmaps show the ATAC-Seq signal intensities surrounding the TSSs of genes in control and adriamycin-treated cells. Adriamycin induced a general loss of chromatin accessibility surrounding TSSs. **c** Adriamycin-specific gene repression was correlated with the loss of chromatin accessibility surrounding regions.

exhibited marginal changes in insulation scores, suggesting that these "gained" boundaries upon adriamycin treatment may correspond to TAD boundaries that are detected at slightly shifted positions (Fig. 5e, f). Collectively, these analyses demonstrated that adriamycin-treatment causes global weakening of TAD boundaries throughout the genome. Interestingly, the differentially expressed genes are more enriched in the vicinity of the TAD boundaries exhibiting the greatest insulation increases (Supplementary Fig. 9b, c, Group1) compared to the TAD boundaries with the least insulation increases (Supplementary Fig. 9b, c, Group2), suggesting the changes in TAD organization may partially contribute to the gene expression changes by altering chromatin interaction landscapes.

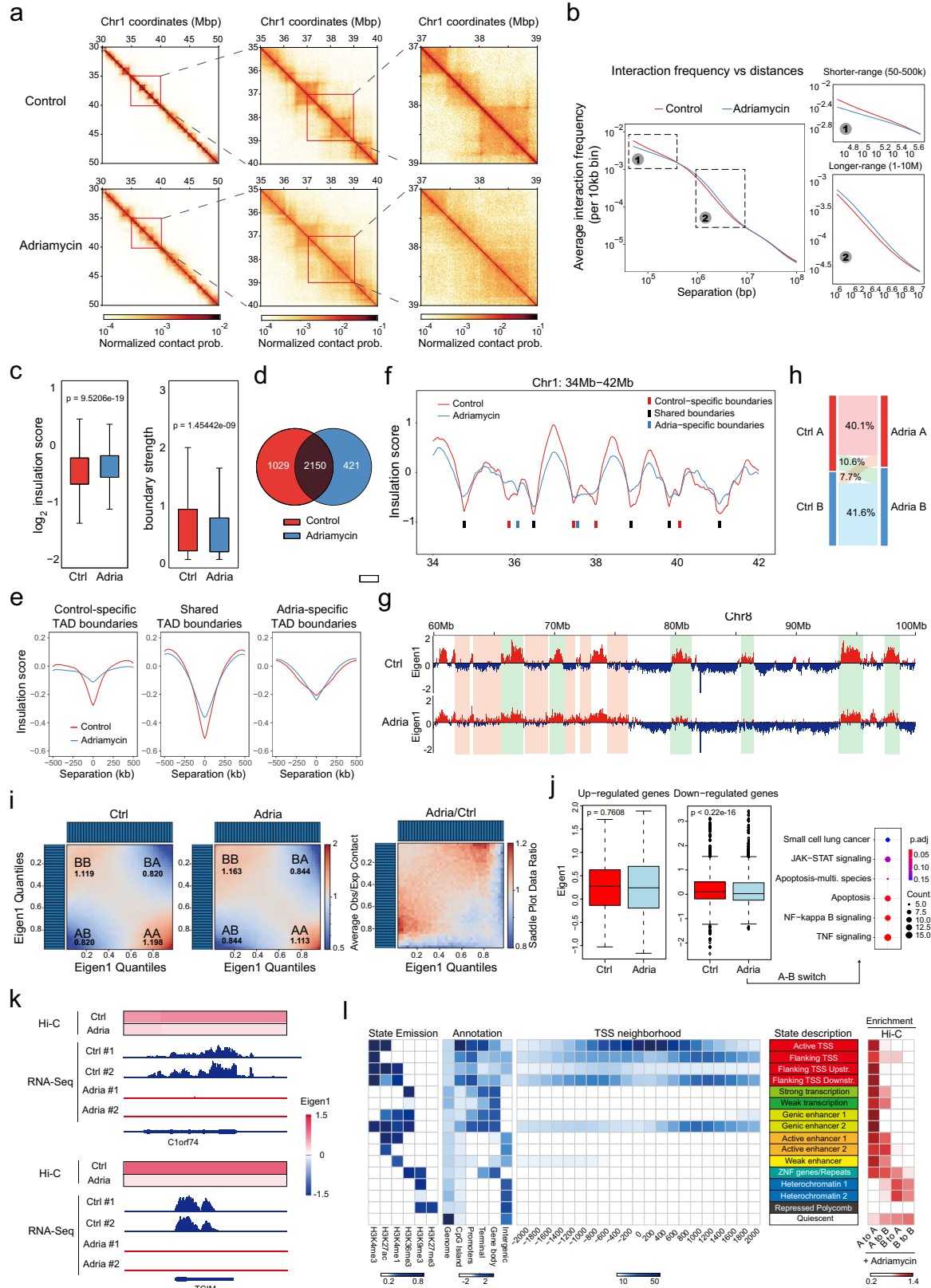

Chromatin is organized into A and B compartments[57]. The A compartments represent active, euchromatic regions, while the B compartments indicate inactive, heterochromatic ones[57,58]. We quantified the patterns of A/B compartment organization in control and adriamycin-treated cells by performing the principal component analysis (PCA) (Fig. 5g, h). Treatment of adriamycin led to substantial changes in A/B compartment organization, with 10.6% (28,791 foci) and 7.7% (20,880 foci) of genomic bins switching from A to B compartment or B to A compartment, respectively (Fig. 5h). We further quantified the strength of compartmentalization by calculating the average chromatin contact frequencies within the same compartment or between different compartments. Upon adriamycin treatment, the

**Fig. 5 | Adriamycin induces genome-wide 3D chromatin conformational change. a** Hi-C heatmaps binned at 10 kb show chromatin interactions patterns of representative regions on Chromosome 1 in control and adriamycin-treated U2OS cells. Multiple TADs and loops are diminished in adriamycin-treated cells. Prob. probability. **b** P(s) curves indicate relationships between chromatin contact probability and genomic distances for chromatin interactions on autosomes in control and adriamycin-treated cells. **c** Box plots quantify the insulation scores and boundary strength for TAD boundaries in control (ctrl) and adriamycin (adria)-treated cells. Boxes, middle 50% of TAD boundary strength. Center bars, medians of boundary strength. Whiskers, 1.5× interquartile range. *p* values are calculated from one-tailed Mann–Whitney *U*-test. *n* = 2 replicates (independently cultured cells were harvested, processed and sequenced separately) for each condition. **d** Venn diagram depicts the overlap between TAD boundaries identified in control (red) and adriamycin-treated (blue) cells. **e** Averaged insulation profiles in control (red) and adriamycin-treated (blue) cells for 1 Mb genomic regions centered at the 10 kb genomic bins containing control-specific (left), shared (middle), or adria-specific TAD boundaries (right). **f** Insulation profiles for a representative genomic region in control (red) and adriamycin-treated (blue) cells. Bars below insulation profiles indicate control-specific (red), shared (black), or adria-specific (blue) TAD boundaries identified in this region. **g–i** An example region (**g**) and overall statistic (**h**) demonstrated the switch of compartment A/B upon adriamycin treatment. **i** The change of contact frequencies between compartments upon adriamycin treatment. Genomic regions belong to the B compartment (B–B) exhibited a notable increase. **j, k** Relationship between gene expression and genomic A/B compartments. **j** Left, the boxplots showing E1 scores of all differentially expressed genes. Boxes, middle 50% of E1 scores. Center bars, medians of E1. Whiskers, 1.5× of inter-quartile range. Right, KEGG analysis of the pathways enriched in down-regulated genes with decreased E1. *p* values were calculated by two-sided Wilcoxon test. *n* = 2 technical replicates for each condition. Examples were showed in (**k**). **l** Histone modification-based learning and annotation of compositions of A/B compartments in U2OS cells performed by ChromHMM analysis (see "Methods" for the detail). The correlations between compartment switches and different chromatin states were analyzed and shown in the heatmap (right).

contact frequencies between genomic regions belonging to the B compartment (B-B) exhibited a notable increase (Fig. 5i). Thus, the Hi-C analyses suggest that adriamycin led to both the expansion and the strengthening of the heterochromatin-rich B compartment, in line with the microscopic observations that adriamycin induced the formation of heterochromatin-enriched chromatin condensates.

We further examined the correlation between the compartment switch and gene expression changes. The genomic regions surrounding down-regulated genes (with log$_2$Foldchange < −2 and Padj <0.05), but not up-regulated genes, were associated with A-B compartment switching, displaying lower Eigen1 scores compared with the control group (*p* value < 2e−16, Fig. 5j and Supplementary Data 3, also Fig. 5k for examples). Some of those down-regulated genes associated with A−B switch were particularly enriched in apoptosis, TNF, and NF-kB pathways (Fig. 5j). Those results suggest that adriamycin-mediated effects on higher-order chromatin conformation are more prone to transcriptional repression regardless of A or B compartments.

To gain more insights into the composition of A/B compartments altered under adriamycin treatment, we performed ChromHMM analysis to annotate the chromatin states[59]. We annotated 16 chromatin states based on public ChIP-Seq datasets of six histone modifications performed in U2OS cells (Fig. 5l), and the annotated chromatin states were highly consistent with annotation of IMR90 fibroblasts based on six histone marks, genomic annotations, and Refseq TSS neighborhoods[59]. Upon adriamycin treatment, some regions of active/flanking TSSs, strong/weak transcription, and active/weak enhancers were switched from A to B compartment, which could result in transcriptional repression (Fig. 5l). Similarly, some heterochromatin regions were switched from B to A compartment upon adriamycin treatment, which could lead to transcriptional activation (Fig. 5l). Notably, genic enhancers and TSS downstream regions, frequently located within gene bodies, were devoid of A to B switch compared to other active enhancers (Fig. 5l). This is in line with the result that the loss of chromatin accessibility was much more pronounced at TSS sites and suggests that the potential selectivity of A/B compartment switch.

**Adriamycin suppresses transposable elements**

Transposable elements (TEs), the mobile DNA elements which make up ~40% of the mammalian genome, are known to be silenced by heterochromatin[60]. As adriamycin−DNA condensates enriched with heterochromatin markers (Fig. 3b), we examined whether the expression of TEs was repressed by adriamycin. Indeed, adriamycin treatment suppressed ~50 TE species (15,661 loci) whose expression was unaffected by etoposide (Fig. 6a–c). Those suppressed TEs are majorly (~75%) long terminal repeat (LTR)-containing endogenous retroviruses (ERVs) but also contain satellites, LINEs, and DNA transposons (Fig. 6d, e and Supplementary Data 4).

We next hypothesized that suppressed TEs were embedded within adriamycin−DNA condensates, leading to the transcriptional silencing. Because of the complexity of multiple alignment for repeat sequences in second-generation sequencing data, it is difficult to precisely allocate each TE. However, we did observe the repression of chimeric long non-coding RNAs or chimeric TEs, which contain unique sequences for mapping, was correlated with the switch of the A/B compartment (Fig. 6f, g). Those results suggest that adriamycin silenced TEs through, at least in part, the redistribution of active and repressed regions in chromatin regions at the megabase scale.

**Adriamycin induces protein-dependent chromatin condensation in vitro**

To investigate the molecular mechanism of chromatin condensation induced by adriamycin, we first tested whether adriamycin-induced chromatin condensation could be recapitulated in vitro. Native chromatin fragments (chromatin fraction) containing both DNA and chromatin-associated proteins were isolated by micrococcal nuclease digestion, and their DNA fraction was further purified by protease K treatment followed by DNA extraction (Fig. 7a). Adriamycin did not lead to condensation of the native chromatin DNA fraction, indicating that the DNA intercalation activity of adriamycin did not contribute to the formation of condensates, at least in vitro (Fig. 7b). As positive control, the DNA fraction of native chromatin formed condensates in the presence of Hoechst 33342 (Fig. 7b). Adriamycin induced the formation of ~1–2 μm fibrous condensates of the chromatin fraction in the absence of the aggregation promoter, magnesium (Fig. 7c and Supplementary Movie 3). Those results indicate that adriamycin-induced chromatin condensation is dependent on chromatin-associated proteins.

We next employed surface plasmon resonance (SPR) to examine the affinities between adriamycin and chromatin-associated proteins/nucleosomes. Among them, adriamycin interacts with histone H1 with an affinity (KD = $1.97 \times 10^{-6}$ M) ~10 times higher than mononucleosomes, which barely contain histone H1 (KD = $1.5 \times 10^{-5}$ M) (Fig. 7c), suggesting that histone H1 is the primary molecular target of adriamycin mediating the observed conformational changes in chromatin. To test if adriamycin could induce phase transition of H1 in a manner similar to native chromatin, we performed an in vitro aggregation experiment with adriamycin and H1-CFP (Fig. 7e). Adriamycin induced the formation of fibrous condensates of H1 by complexing with it (Fig. 7e and Supplementary Movie 4). Interestingly, in contrast to the "fusion" of the liquid-phased H1, the "fusion" of those fibrous adriamycin-H1 condensates seems to be mediated by their size expansion (Fig. 7f). In addition, the FRAP experiment also revealed a slow fluorescence recovery rate of adriamycin (Fig. 7g and Supplementary Movie 5). Together, those results suggest a plausible viscoelastic or gel/solid-like property of adriamycin−H1 condensates.

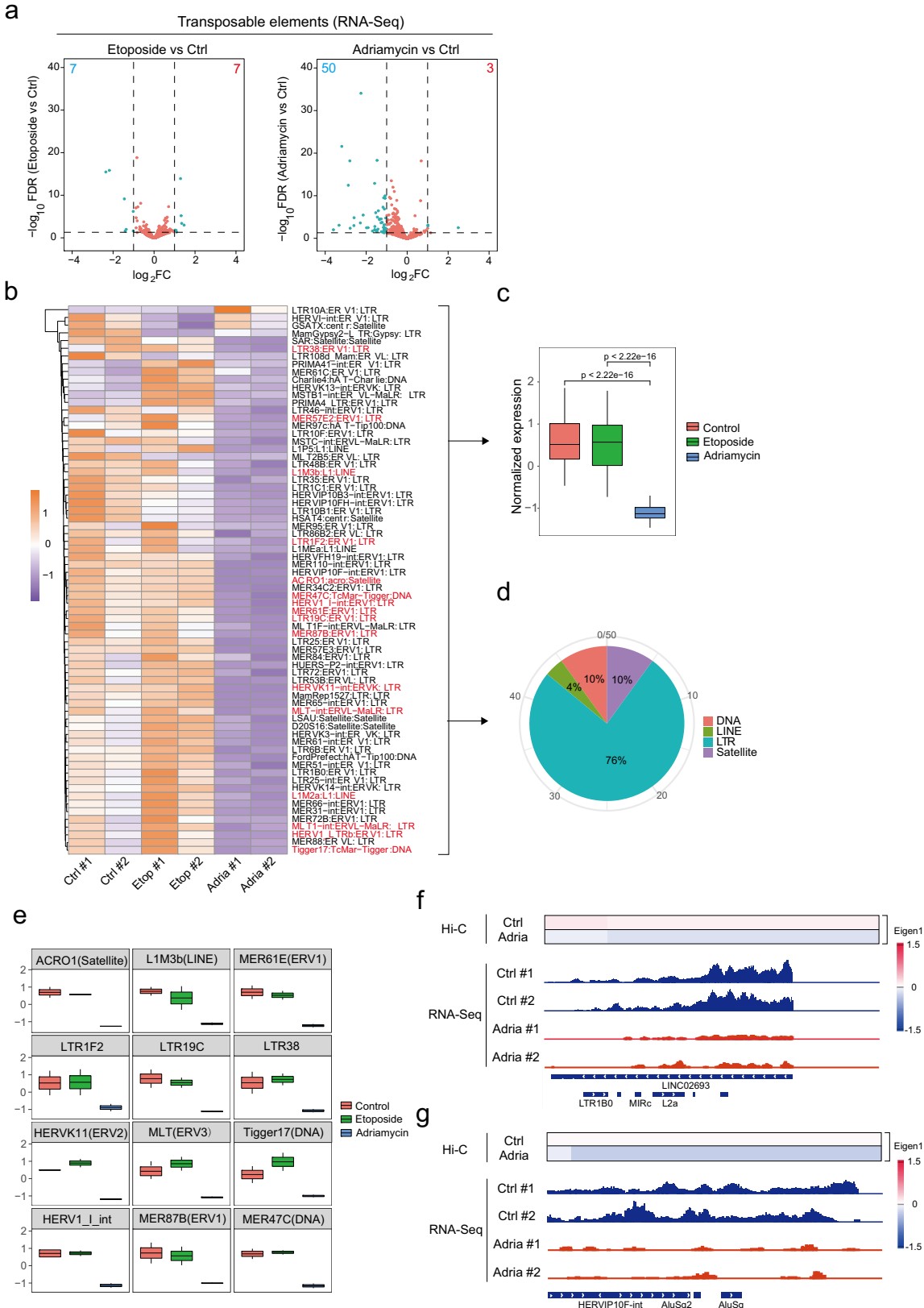

Next, we performed immunostaining to trace the interaction between adriamycin and histone H1 in vivo, and observed the co-localization of adriamycin, histone H1, and DNA in condensates gradually formed over time (Fig. 7h). To gain more insights into the mechanism, we dissected H1 into three fragments with structural prediction and purified fusion proteins (Supplementary Fig. 10a). We found that the C-terminal disordered fragment exhibited the highest affinity for adriamycin (Supplementary Fig. 10b) and the ability to form phase-separated condensates with adriamycin in vitro (Supplementary Fig. 10c). Together, those results suggest that the formation of adriamycin-chromatin condensates could be mediated by inducing the phase separation of H1 through interacting with its C-terminal region.

**Fig. 6 | Adriamycin treatment leads to suppression of TE (transposable elements) expression. a** Volcano plots show differentially regulated TE transcripts after etoposide and adriamycin treatment. TE transcripts with abs(log$_2$(foldchange)) >1, Padj <0.05 were highlighted by cyan color. **b** Heatmap showing relative expression of differentially regulated TE transcripts in each treatment groups. **c** Normalized expression of top 50 adriamycin induced downregulated TE transcripts in each treatment groups. The boxplot shows the normalized gene expression (read counts) in control, etoposide-, and adriamycin-treated conditions. Boxes, middle 50% of normalized gene expression. Center bars, medians of normalized gene expression. Whiskers, 1.5× of inter-quartile range. *p*

values were calculated by two-sided Wilcoxon test. *n* = 2 technical replicates for each condition. **d** Pie chart showing species classification of top 50 adriamycin induced downregulated TE transcripts. **e** Relative expression of selective 12 TE transcripts downregulated upon adriamycin treatment. The boxplots show the normalized expression (read counts) of TEs in control, etoposide-, and adriamycin-treated conditions. Boxes, middle 50% of normalized gene expression. Center bars, medians of normalized gene expression. Whiskers, 1.5× of inter-quartile range. **f, g** Examples of TE-containing non-coding RNAs (**f**) and TEs (**g**) demonstrate the correlation between compartment A to B transition and expression repression induced by adriamycin. Ctrl control, adria adriamycin.

We proposed a model illustrating the chromatin condensation/re-organization process induced by adriamycin (Fig. 7i). After entering the nucleus, adriamycin interacts with primarily heterochromatin but also euchromatin, at least for enhancer regions, rendering the conformation of both by complexing with histone H1, leading to the phase transition of histone H1 and chromatin. This phase transition results in the condensation of chromatin, weakened TAD boundaries throughout the genome, and transcriptional repression of coding genes and TEs. Overall, this model proposes an action mechanism for Adriamycin.

## Discussion

Adriamycin (doxorubicin) is one of the most effective and widely used drugs for treating various adult and pediatric cancers in the clinic. However, chemotherapies using adriamycin have potential side effects, especially cardiomyopathy and congestive heart failure[61–64]. The underlying mechanisms of cytotoxicity against tumor cells and/or cardiomyocytes are still unclear. This complexity is, at least in part, due to adriamycin's unique feature as both an anthraquinone, which can mediate redox cycling, and a DNA intercalator, which poisons DNA topoisomerase II and results in DNA damage[65–67]. On the one hand, ROS generated by adriamycin has been attributed to its cardiotoxicity[68]. On the other hand, ROS scavengers failed to exhibit effective protective effects against adriamycin-induced myocardiopathy[69,70]. In addition, adriamycin-induced cytotoxicity in MEFs is dependent on Top2β[53], and ablation of Top2β in mouse cardiomyocytes protects mice from progressive heart failures induced by adriamycin[71], suggesting that adriamycin-induced cardiotoxicity is mediated by its DNA intercalation/TOP2-poisoning activity. Multiple mechanisms were also proposed for adriamycin-induced cytotoxicity, including apoptosis, ferroptosis, autophagy, epigenetic alterations, and mitochondria-mediated apoptosis[72].

In the present study, we demonstrate an activity of adriamycin in inducing global, higher-order chromatin conformation changes by promoting phase transition of chromatin. Hi-C data suggests the strengthening and expansion of the heterochromatic B compartment, consistent with the enriched heterochromatin markers revealed by microscopic observations. Previous studies showed that the liquid–liquid phase separation property of HP1 could drive heterochromatin formation and the genome compartmentalization[20,73,74]. Adriamycin could similarly enhance heterochromatin formation by promoting phase separation, thereby affecting genome compartmentalization. Notably, we also show that adriamycin caused a global weakening of TAD organization. Such an effect may be independent of the heterochromatin-related condensate formation, as TAD formation is driven by dynamic cohesin/CTCF-mediated loop extrusion[75]. The mechanism by which adriamycin affects the binding and processivity of cohesin/CTCF remains unclear and awaits further elucidation.

Histone H1 plays a crucial role in gene silencing by modulating chromatin compaction and 3D genome organization in vivo[76]. In vitro, the disordered C-terminal domain of H1 has been shown to complex with DNA and promote the condensation of chromatin[17,77]. Thus, both in vitro and in vivo experiments indicate that the 3D genome organization activity of H1 could be link to its ability to induce phase

transition. Our experiments further demonstrated that even in the absence of DNA, adriamycin could promote the phase transition of H1 to form adriamycin-H1 fibrous condensates, indicating that the predominant activity of adriamycin to reorganize 3D chromatin conformation could be through adriamycin-H1 interaction. It's also possible that adriamycin interacts with other histones or chromatin-associated proteins and induces phase transitions.

There are seemly contradictory reports regarding the relationship between H1, adriamycin, and DNA. On the one hand, adriamycin has been shown to interact with H1 with an affinity higher than other histones[77,78], consistent with our SPR analysis. On the other hand, adriamycin could evict or displaces nucleosome histones or H1 from DNA by competing with histones for space in minor grooves[79,80], indicating that adriamycin has a higher affinity for DNA than H1 in this scenario. We think these two observations might not be mutually exclusive and could happen simultaneously or sequentially. One possible explanation is that adriamycin could initially displace or evict H1 from chromatin, and the displaced or free H1 could form the condensates with adriamycin and re-associate with chromatin, leading to the re-organization of its 3D conformation (Fig. 7g). Alternatively, adriamycin could directly associate with DNA-bound H1 in a site-specific manner (Fig. 7g). The real dynamics of histones/nucleosomes influenced by adriamycin still await further investigation.

Finally, global reorganization of 3D genome conformation happens during biological processes, such as stem cell differentiation. During those processes, the "rigid" DNA fraction of the heterochromatin is reorganized. Yet, approaches to studying the correlation between the physical property of chromatin, the higher-ordered structure, and their influence on gene expression are still limited. The distinct morphology and property of the fibrous condensates formed by adriamycin and chromatin may reflect a viscoelastic property in contrast to canonical liquid–liquid phase separation molecules. Investigating of these unusual condensates will enable us to connect the physical property of chromatin to its conformation and biological consequences. Further modification or screening for this type of chemicals could facilitate basic research or clinical applications.

## Methods

This research complies with all relevant ethical regulations of Shanghai Institutional Animal Care and Use Committee (IACUC) guidelines and under an approved IACUC protocol of ShanghaiTech University.

### Cell culture

U2OS (SCSP-5030), HCT116 (SCSP-5076), and HeLa (SCSP-504) cells were purchased from the National Collection of Authenticated Cell Cultures, China. All cell lines were validated by corresponding STR identifiers by the cell bank. MEF, U2OS, HCT116, HeLa, and human mesenchymal stem cells were cultured in DMEM (Gibco, C11965500CP) supplemented with 10% fetal bovine serum (FBS) (Lonsera, S711-001S) and 1% penicillin/streptomycin (Thermo, 15140122). Cells were cultured at 37 °C under 5% $CO_2$ in the air. Primary cardiomyocytes were cultured in DMEM supplemented with 10% FBS and 1% penicillin/streptomycin. Primary hepatocytes were cultured in human fibroblast medium (HFM).

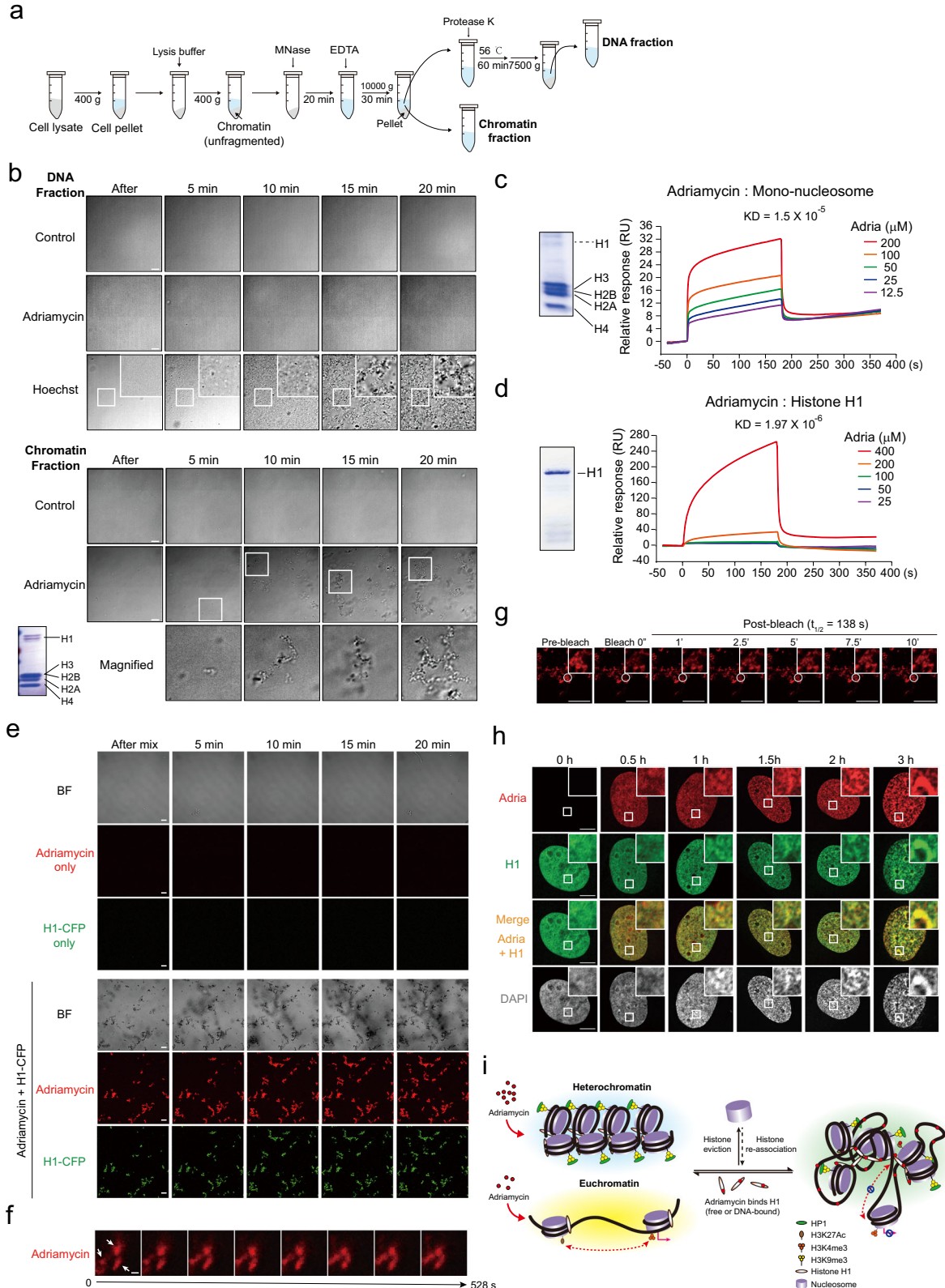

For the drug treatment, cells were treated with drugs at the following concentration unless specifically annotated: adriamycin (ABCONE, D77185), 1.5 μg/ml; etoposide (ABCONE, E01505), 100 μM; camptothecin (meilunbio, MB1044), 10 μM; bleomycin (TargetMol, T6116), 60 μg/ml; hydrogen peroxide (Greagent, G82427B), 100 μM; psoralen (MCE, HY-N0053), 450 μM; cisplatin (Meilunbio, MB1055), 10 μM; cytophosphane (MCE, HY-17420), 40 μM; dactinomycin (MCE, HY-17559), 2 μM daunorubicin (MCE, HY-13062); 5 μM aclarubicin (ENZO, BML-AW8655-0005); 0.5 μg/ml; erastin (Stemcell, 100-0545), 4 μM; trichostatin A (TSA) (MCE, HY-15144), 100 nM; 5,6-dichlorobenzimidazole riboside (DRB) (Sigma, D1916), 20 μM for 24 h; THZ1 (MCE, HY-80013), 500 μM. z-VAD-fmk (Sigma-Aldrich, V116), 20 μM;

**Fig. 7 | Adriamycin induces phase transition of chromatin and histone H1 in vitro. a** Isolation of native chromatin and further purification of its DNA fraction. **b** Adriamycin did not induce aggregation of the DNA fraction of chromatin, but 0.1 mg/ml Hoechst 33342 did. In contrast, adriamycin induced the phase transition of native chromatin which contains the linker histone H1 and nucleosome histones, suggesting this phase separation phenomenon is dependent on chromatin-associated proteins. Scale bar, 10 μm. The experiment was repeated at least three times with similar results. **c, d** SPR analysis revealed the affinities between adriamycin (**c**) and mono-nucleosomes in which H1 is absent (**d**). Experiment of b and c were repeated twice with similar results. **e** In vitro condensation of adriamycin and H1-CFP. By itself, neither adriamycin nor H1-CFP formed condensate. Mixing adriamycin and H1-CFP led to the phase separation, forming condensates containing both adriamycin and H1-CFP. Scale bar, 10 μm. BF, bright field. **f** The "expansion–fusion" dynamics of adriamycin condensates. Also see the text for the description. Scale bar, 1 μm. **g** FRAP experiment demonstrated the diffusible property of adriamycin within adriamycin-H1-CFP condensates. Scale bar, 10 μm. **h** The co-localization of adriamycin, histone H1 and DNA in cells. U2OS cells were treated with 1.5 μg/ml adriamycin (adria), fixed at indicated time points, and stained with anti-H1 antibody and DAPI. Scale bar, 10 μm. Source data are provided as a Source data file. **i** Proposed model for adriamycin-induced chromatin reorganization.

necrostatin-1 (Beyotime, SC4359), 10 μM, ferrostatin-1 (Meilunbio, MB4718), 1 μM; MG132 (Meilunbio, MB5137), 0.5 h pre-treatment for 4 μM, dexrazoxane hydrochloride (ICRF-187) (MCE, HY-76201), 3 h pretreatment for 200 μM. Knockdown of *hTop2β* were performed by infecting U2OS cells with lentiviruses producing by pLKO.1 transfer vector encoding *hTop2β* short-hairpin RNA (shRNA). The shRNA oligos used for self-annealing and pLKO.1 insertion (synthesized by GENEWIZ, Shanghai) are: 5'-CCGGGAACTTGGACACAGGTATATACTCGAGTATA-TACCTGTGTCCAAGTTCTTTTTG-3' and 5'-AATTCAAAAAGAACTTGG ACACAGGTATATACTCGAGTATATACCTGTGTCCAAGTTC-3'.

### Animals

Wild-type (WT), 6–8-week-old C57BL/6 mice obtained from Shanghai Model Organisms Center (Shanghai, China) were kept in sterilized filter top cages with 40–60% humidity and a 12 h day/night cycle at 22 °C. Mice were mated in 1:1 ratio and plug-checked for getting P1 neonates (for the primary cell isolation) or E13.5 mouse embryonic fibroblasts (MEFs). All mouse experiments were conducted in accordance with IACUC guidelines and under an approved IACUC protocol of ShanghaiTech University.

### Primary cell culture

Hepatocytes and cardiomyocytes were isolated from C57BL/6J mice aged at postnatal day 1 (P1). For cardiomyocytes, dissected hearts were washed with calcium/magnesium-free Hank's balanced salt solution (CMF-HBSS) (Hyclone, SH30031.01) briefly, followed by incubating in lysis buffer containing collagenase IV (Stemcell, C9263) and 0.25% trypsin (Gibco, 15050065) in HBSS at 37 °C for 10 min. The supernatant was moved to another tube and the tissues were repeatedly digested 5 times until clumps disappeared. After gentle resuspension and filtering through 100 μm cell strainers, cell suspensions were pelleted at $300 \times g$ at room temperature. Cardiomyocytes were resuspended and cultured for 3–5 days before adriamycin treatment. For hepatocytes, dissected livers were washed with CMF-HBSS briefly and treated with 0.5 mg/ml collagenase I (Worthington, LS004194) at 37 °C for 30 min. After gentle resuspension, cells were pelleted by centrifugation at $100 \times g$. Pelleted cells were resuspended and incubated with red blood cell (RBC) lysis buffer (Sigma, R7757) for 10 min, filtered through the 100 μm cell strainer. Hepatocytes were then centrifuged, resuspended and cultured for 8–9 days before adriamycin treatment.

### Synchronization and cell cycle analysis

To unify the cell cycle progression, U2OS cells were synchronized with 9 μM RO-3306 (MCE, HY-12529) for 18 h and subsequently released in fresh culture media. Adriamycin was treated at different time points after release. To analyze the cell cycle, cells were trypsinized and kept on ice in 1 ml of PBS. Cold ethanol was added dropwise to cells, followed by incubation for 30 min on ice. Cells were then centrifuged at $300 \times g$ for 10 min at 4 °C and resuspended in 1 ml of PBS. After two times PBS washing, cells were resuspended in PBS containing 100 μg/ml RNase A (TIANGEN, RT405) and incubated for 30 min at 37 °C. Cells were then centrifuged at $300 \times g$ for 10 min at 4 °C and resuspended in PBS containing 5 μg/ml propidium iodide (Biolegend, 421301) for FACS analysis on BD LSRFortessa.

### Super-resolution microscopy (STED and SIM)

Cells were seeded on glass coverslips coated with 0.1% gelatin, fixed with 4% paraformaldehyde PBS for 10 min, and stained with 0.5 μM SiR–Hoechst (Cytoskeleton, CY-SC007) for 10–12 h. The slides were mounted with prolonging Golden antifade reagent (Life Technologies, P36930). For STED, images of fixed cells were taken on a Leica SP8 3X microscope equipped with a 775 nm STED laser, a 640 nm excitation line, and hyD detection. For SIM, images of live U2OS cells were taken on an Elyra 7 Lattice SIM with the Plan-Apochromat ×63/1.4 Oil DIC M27 objective lens and the PCO edge 4.2 sCMOS camera. Images were reconstructed by the Lattice SIM$^2$ algorithm.

### Immunofluorescence staining and live cell imaging

Cells were seeded on glass coverslips coated with 0.1% gelatin, fixed with 4% paraformaldehyde PBS for 10 min, blocked in buffer containing 2.5% BSA and 0.3% Triton X-100 in PBS for 1 h, and incubated with primary antibodies overnight at 4 °C. After washes, cells were incubated with secondary antibodies and genomic DNA were stained by 0.1 μg/ml Hoechst 33342 (Beyotime, C1022) for 1 h at room temperature. The coverslips were mounted on glass slides in VECTASHIELD antifade mounting medium (Vectorlabs, H-1000-10) and sealed. Cells were imaged with ZEISS 980 Airyscan2. The primary antibodies used are: Lamin A/C (ABclonal, A19524), 1:100; Tnni3 (ABclonal, A6995), 1:150; Albumin (Life Technologies, A90-134A), 1:750; H3K9me3 (ABclonal, A2360), 1:150; H3K4me3 (Abcam, ab8580), 1:100; H3K27ac (PTM, 116), 1:100; H1 (PTM, 6054), 1:100; MED1 (Abcam, ab64965), 1:200, 53BP1 (ABclonal, A5757), 1:200; γ-H2AX (ABclonal, AP0099), 1:200. The secondary antibodies used are Alexa Fluor 647-AffiniPure goat anti-mouse IgG (H + L) (Jackson, 115-605-003) and Alexa Fluor 647-AffiniPure goat anti-rabbit IgG (H + L) (Jackson, 111-605-003) (1:500).

For p53 localization staining, p53-Halotag MEF (will be described somewhere else) was treated with 1.5 μg/ml for 4 h and the intracellular p53 was visualized by treating with 5 nM Janelia Fluor 646 (Tocris, 6148/1) overnight. For live cell imaging, cells were plated on 35 mm glass-bottom dishes pre-coated with 0.1% gelatin and grown until ~50% confluency. Cells were imaged by Leica Thunder Imager.

### Fluorescence recovery after photobleaching (FRAP)

For live cell imaging, cells were plated on the 35-mm glass-bottom dishes and grown typically overnight. Before imaging, cells were treated with 1.5 μg/ml adriamycin for 4 h. FRAP was carried out on the Nikon CSU-W1 system. Images during FRAP were acquired with the 561 nm laser for adriamycin acquisitions, while 488 nm laser was used for bleaching. Images were acquired prior to bleaching a circular area with 3.51 μm$^2$ using 60% laser power for 100 ms, followed by 1 min for monitoring the recovery. Signals were corrected for photobleaching using a similarly sized unbleached area and then normalized to the ratio between the average intensity of the pre-bleach images and the

lowest post-bleach intensity. Averages ± standard deviation (SD) from 10 to 15 cells per condition were plotted.

## Electron microscopy

Cells were harvested by centrifuging at $300 \times g$ for 10 min, followed by fixation with 2.5% glutaraldehyde overnight at 4 °C. Cells were then washed 3 times in PBS and fixed with $OsO_4$. After washes, cells were dehydrated using gradient concentration ethanol (30–100%) and embedded with phenolic epoxy resin overnight. The section was performed on Leica EM UC7 and images were taken on GeminiSEM 460.

## Quantification of condensation

The whole quantification is illustrated in Fig. S3A. Briefly, cell images were transferred to 8-bit by FIJI software. At least three spots were chosen for RDF and L-function calculation and at least 10 cells were counted for each condition. $\gamma$ was set as 0.2 μm.

The RDF is given by the equation:

$$g(r) = \left[ \left( \frac{S}{N-1} \right) \frac{1}{\pi \left( 2r\Delta r + \Delta r^2 \right)} \right] \left[ \frac{1}{N} \sum_{i=1}^{N} \sum_{i \neq j} \delta(r - r_{i,j}) \right] \quad (1)$$

The L-function is given by the equation:

$$L(r) = \sqrt{\frac{K(r)}{\pi}} \quad (2)$$

## Labeling of heterochromatin and euchromatin by dUTP-Cy5

Cells were labeled when they were approximately 80% confluent. The medium was completely removed then added DMEM containing dUTP-Cy5 (1:50 dilution). The cell lawn was scratched with a 26-gauge hypodermic injection needle in parallel lines from one side to the other side. The dish was rotated 90° and the cell was scratched a second time as described above. After 2 min, cells were washed with 1× PBS and incubated in a fresh conditioned medium for 24 h before imaging.

## Native chromatin fragment preparation

Approximately 6 million U2OS cells were lysed in lysis buffer (10 mM Tris-Cl, pH7.5, 10 mM NaCl, 3 mM $MgCl_2$, 10 mM sodium butyrate, 250 mM sucrose and 0.25% V/V of NP-40). Nuclei were washed two times with the same lysis buffer and were collected by centrifugation at 400 g for 10 min. Nuclei were resuspended in MNase digestion buffer (15 mM Tris–Cl pH 7.5, 15 mM NaCl, 250 mM sucrose, 2 mM $CaCl_2$, 60 mM KCl, 15 mM β-mercaptoethanol, 0.5 mM spermidine, 0.15 mM spermine, 0.2 mM PMSF, protease and phosphatase inhibitors) and were digested with 25 U/ml of micrococcal nuclease at 37 °C for 20 min. The reaction was stopped by the addition of EGTA to 10 mM and nuclei were collected by centrifugation at $400 \times g$ for 10 min. Nuclei were next resuspended in 10 mM EDTA for 30 min on ice, which resulted in nuclear lysis and the release of chromatin fragments into the medium. The EDTA soluble chromatin was separated from insoluble nuclear material by centrifugation at $10,000 \times g$ for 15 min. The isolated soluble chromatin was dialyzed overnight against 1 mM Tris-Cl (pH 8.0) and 0.1 mM EDTA at 4 °C. With this protocol and USOS cell number, the resulting native chromatin concentration was usually ~1.5 mg/ml. To isolate the DNA fraction from native chromatin by treating chromatin with 0.1 mg/ml Proteinase K at 56 °C for 60 min. Then centrifugation at $750 \times g$ for 10 min, the DNA fraction in the supernatant was. With this protocol and USOS cell number, the resulting DNA concentration was usually 2.8–2.9 $OD_{260}$.

## In vitro condensation assay

Chromatin condensates were formed by first preparing the buffer solution containing 1 mM Tris-Cl, pH 7.5. 18 μl of purified chromatin was spotted on 35-mm glass-bottom dishes and mixed with 2 μl buffer solution to make the final concentration 5 mM $MgCl_2$ or 1.5 mg/ml adriamycin. Drops were imaged by Leica Thunder Imager immediately. Time-lapse images were collected every 30 s until 20 min.

## Imaging and FRAP of aggregates from in vitro experiments

Droplets of in vitro experiments were imaged every 8.7 s for one frame at ×60 on Zeiss LSM 980 Airyscan2. For the FRAP experiments, approximately 4 μm² was bleached with the 514 nm laser with 100% power of the quantifiable laser module (QLM) and the recovery was observed at 1% power for 20 min.

## Surface plasmon resonance (SPR)

SPR experiments were performed on Biacore 8 K (GE Healthcare). All assays were performed with a running buffer containing 10 mM HEPES pH 7.4, 150 mM NaCl, 3 mM EDTA and 0.01% v/v Tween-20 at 25 °C. Recombinant histone H1 or native mono-nucleosomes (purified by Active Motif nucleosome preparation kit, 53504) were immobilized to a single flow cell on a CM5 sensor chip (GE Healthcare). Three samples containing only running buffer were injected over both sample and reference flow cells, followed by 2-fold serial dilutions of purified drugs (30 μl/min, association 180 s, dissociation 180 s). To measure the binding affinity of adriamycin to H1 or mono-nucleosomes, serial dilutions of adriamycin were flowed over immobilized H1 or mono-nucleosomes. All the binding data were double referenced by blank cycle and reference flow cell subtraction. The resulting sensorgrams were fit to a 1:1 Langmuir binding model using the Biacore Insight Evaluation Software (GE Healthcare). The data were processed and analyzed using Biacore 8 K Evaluation Software Version 3.0 (Cytiva, Marlborough, MA, USA). The responses recorded on the FC1 were subtracted from those in the corresponding FC2. The responses from the nearest buffer blank injection were subtracted from the reference subtracted data (FC2-FC1) to yield double-referenced data.

## Transcriptome analysis

The libraries were then sequenced by the Illumina NovaSeq 6000, and the pair-ended reads of 150 bp were generated. The reference genomes (Homo sapiens GRCh38) and the annotation file were downloaded from the ENSEMBL database (http://www.ensembl.org/index.html). HISAT2 v2.1.0 was used for building the genome index, and clean data was then aligned to the reference genome. The read count for each gene in each sample was counted by Subread v2.0.0 featureCounts function.

Differentially expressed genes in Supplementary Data 1–3 were identified using DESeq2 v1.30.1. Two-tailed $p$ values were calculated by Wald test and the resulting $p$ values were adjusted using Benjamini and Hochberg's approach (Padj) for controlling the false discovery rate. Padj < 0.01 and $|log_2(foldchange)| > 1$ were set as the threshold for significantly differential expression. Genes were ranked according to the degree of differential expression in the two samples by fcros v1.6.1[81]. Hallmark gene sets were downloaded from Molecular Signatures Database (http://www.gsea-msigdb.org/gsea/msigdb). We use clusterProfiler v3.18.1[82] to do the GSEA analysis.

## TEs discovery and abundance estimation

Trimmed clean RNA-Seq data were mapped by STAR with −winAnchorMultimapNmax 200 and --outFilterMultimapNmax 100 parameters to allow the recovery of multi-mappers. Next, TEtranscripts software[83] was used to estimate gene/TE abundances and conduct differential expression analysis. GTF file of transposable element annotations was downloaded from https://hammelllab.labsites.cshl.edu/software/#TEtranscripts. R package DEseq2 was used to perform a pairwise comparison between each treatment group and calculate differentially expressed TEs.

## Hi-C data analysis

Mapping and heatmap generation. Raw reads were firstly trimmed using TrimGalore (version 0.6.7) with default settings. The data were then processed using a standard Hi-C processing pipeline recommended by the 4D Nucleome Data Portal (Reiff, Schroeder, et al.[84]) (https://data.4dnucleome.org/resources/data-analysis/hic-processing-pipeline). Briefly, the trimmed reads were first mapped to the human genome (GRCh38) using BWA MEM (version 0.7.17-r1188) with the -SP5M option. The mapped reads were then parsed using pairtools (version 0.3.0). After data aggregation and normalization, contact matrices in the.cool file format were generated using the cooler package (Abdennur and Mirny 2020) (version 0.8.11). Hi-C heatmaps at selected regions were generated using the cooltools package (version 0.5.0).

For A/B compartment analysis, Eigen value decomposition was performed on cooler matrices binned at 10 kb resolution using the call-compartments utility from the cooltools package (version 0.5.0). The A/B compartment was defined using the first eigenvector values.

TAD boundary analysis. Insulation analysis and TAD calling were performed on cooler matrices binned at 10 kb resolution using the diamond-insulation utility from the cooltools. TAD boundaries with boundary strength above 0.1 were considered high-confidence boundaries and used in subsequent analysis. When comparing the locations of TAD boundaries between untreated and Adriamycin-treated U2OS cells, the 10 kb bin for each TAD boundary was extended by 50 kb upstream and downstream to make a 110 kb boundary zone. The TAD boundary zones that were partially overlapping were considered common TAD boundaries in untreated and adriamycin-treated U2OS cells.

## ATAC-Seq data analysis

For all ATAC-Seq datasets, raw reads were firstly trimmed using TrimGalore (version 0.6.7) with default settings and then mapped to the human genome (GRCh38) with bowtie2[85] (version 2.3.5, --very-sensitive). PCR duplicates and mitochondrial reads were excluded using samtools (version 1.9). To correct bias caused by Tn5 transposase, all mapped reads were offset by +4 bp for the + strand and −5bp for the − strand using the alignmentSieve −ATACshift function in the deeptools package[86] (version 3.5.0). ATAC-Seq peaks were identified using MACS2[87] (-f BAMPE, version 2.2.7.1). Genome-wide differential ATAC-Seq peaks were identified using the DiffBind package (https://bioconductor.org/packages/release/bioc/html/DiffBind.html) (version 3.6.1) based on edgeR analysis [FDR < 0.05, abs(log2FC)>1]. Peaks annotation was performed using ChIPseeker[88] (version 1.30.2).

## ChIP-Seq data processing and ChromHMM analysis

ChIP-seq data of six histone marks H3K4me3, H3K4me1, H3K27ac, H3K36me3, H3K9me3 and H3K27me3 were download from GEO database with accession numbers GSE141139 (H3K4me3, H3K27ac, and H3K4me1), GSE31755 (H3K36me3 and H3K9me3), and GSE130230 (H3K27me3) and converted to fastq format using fasterq-dump of SRA Toolkit3.0.3. For the processing of ChIP-seq data, fastq raw reads were cleaned by the fastp software (version 0.3.1) (https://github.com/OpenGene/fastp), then the trimmed reads were mapped to UCSC human hg38 genome using bowtie2 (version 2.2.9). Reads were sorted by SAMtools (version 1.6) into bam files for the following ChromHMM analysis. To evaluate the chromatin states in U2OS cells, we performed ChromHMM (version 1.24) analysis to characterize the switches of chromatin before/after Adriamycin treatment. Bam files were binarized at 200-bp resolution by BinarizeBam program and chromatin states were learned in an 18-state model. The fold enrichment of each state for genomic elements, TSS neighborhood, and A/B compartments were conducted by the "OverlapEnrichment" program of ChromHMM software[59].

## Reporting summary

Further information on research design is available in the Nature Portfolio Reporting Summary linked to this article.

## Data availability

The data that support this study are available from the corresponding authors upon request. The raw sequencing data of ATAC-Seq, RNA-Seq, and Hi-C generated from this study have been deposited to the GEO database under the accession codes GSE222220, GSE222221, and GSE222637. Source data are provided with this paper.

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

## Acknowledgements

We thank Teng Feng, Junjin Zhu, Shiyou Jiang, Xiaohui He, and Min Diao for technical supports, as well as Dr. Xiong Ji, Dr. Fang Bai, Dr. Hui Zhang, Dr. Guisheng Zhong, and Dr. Guanjun Gao for their critical reading. The research was supported by National Key R&D Program of China (2020YFA0710800, C.L.), National Natural Science Foundation of China (31871487, C.L.), and the ShanghaiTech University start-up fund. We thank the Multi-Omics Core Facility (MOCF), Molecular Imaging Core Facility (MICF), and Molecular and Cell Biology Core Facility (MCBCF) at the School of Life Science and Technology, ShanghaiTech University for providing technical support. We thank Yi Zhang from the Discovery Technology Platform at the Shanghai Institute for Advanced Immuno-chemical Studies (SIAIS), ShanghaiTech University for providing technical support of Biacore8K. We would like to especially thank Xiaoming Li, Ziwei Yang, Rui Wang, and Chengyu Fan for expert advice on microscopy. We apologize for critical works that are not cited due to space constraints.

## Author contributions

T.W. and S.S. designed and performed microscopic, RNA-Seq, and in vitro experiments; Y.S. and P.J. performed and analyzed Hi-C and ATAC-Seq experiments; G.H. and G.F. performed the TE and multi-variant analysis; Q.Y. analyzed the RNA-Seq results; Z.S. and Z.L. synthesized Texas Red-cisplatin; H.M. assisted heterochromatin assays; A.C. and C.W. helped the isolation of primary cardiomyocytes and hepatocytes; K.Y. and S.Z. analyzed the structure of histone H1; C.L. and Q.B. designed and performed research, analyzed data, and wrote the manuscript.

## Competing interests

The authors declare no competing interests.
