## [Peer Review File · Nature Communications]

Chemical-induced phase transition and global conformational reorganization of chromatinREVIEWER COMMENTS

Reviewer #1 (Remarks to the Author):

The study applied a small library of molecules targeting genomic DNA to examine the physical state of (viscoelastic) chromatin by STED and electron microscopy. They found that Adriamycin induced spatially resolved DNA puncta – preferentially affecting euchromatin, and independently of Adriamycin effects on ROS, ferroptosis, the TOP2-DNA ligase, or cell death. Puncta were induced in a cell type specific way (cardiac myocytes but not hepatocytes), and they were reversible after removal of agonist - suggesting physiologic relevance. In affected cells, adriamycin was found to colocalize exactly with the DNA puncta implicating direct interaction with components of the puncta – and this enabled studies showing the Adriamycin-induced puncta had features of liquid-like biomolecular condensates - unlike those reported before. Notably, the formation of adriamycin-condensates was closely correlated with structural changes in organization and accessibility of chromatin – and gene expression.

The evidence for effects on chromatin structure and function appears to be strong – though it should be reviewed by someone working in the field, which I am not. I do find the evidence for condensate definitive – features of liquid-like behavior are convincingly demonstrated; and the condensates are reproduced in vitro with known components. The condensates are shown to result mechanistically by direct interaction with chromatin (not DNA) – by interaction with H1 histone specifically. This, and the close correlation between condensate formation and chromatin structure and function, are meaningful contributions to the field.

Overall, the studies are well conceived, technically well done and well controlled, and they are reasonably interpreted. Though why condensate formation may be important to regulation of chromatin structure is not made clear. These ideas might be discussed.

Line 112 – use of word “superb” is commentary – perhaps substitute “high-resolution”

Line 151 – use of term “condensates” here (and following) not yet justified – evidence for condensate formation is not yet presented until Fig 2 and supplemental data.

Line 248-249 – and 258-260 – sentences need rewording – hard to understand.

Line 397 – figure reference is confusing.

Reviewer #2 (Remarks to the Author):

The manuscript by Shi et al. reports that adriamycin induces the condensation of chromatin and influences the 3D genome organization. They screened out adriamycin from different types of DNA targeting chemicals as it induces chromatin condensation which is dynamic and reversible, co-localized with heterochromatin markers. Interestingly, this change is independent of its TOP2 poisoning activity, DNA damage, ROS, and programmed cell death. In addition, after Adriamycin treatment, the loss of TAD boundaries, decrease of chromatin accessibilities and changes of gene expression were revealed by genomic data such as Hi-C and ATAC-seq. The authors further indicated that adriamycin interacts with histone H1 and induces phase transition. Overall, this is an interesting and comprehensive study and the manuscript was well-written. Below are some suggestions which could help the authors to improve their manuscript.

1. The mechanism on how Adriamycin regulate chromatin structure remain unclear. The working model (Fig 7G) isn't well supported by Fig 7, which presents mainly in-vitro data. How about the interaction of Adriamycin and histone H1 in live cells? The adding of Adriamycin lead to the formation of H1 condensates in cells? Are those potential H1 condensates associated with the puncta as observed in Fig 1?
2. The authors stated that “...adriamycin induced drastic chromatin condensation in

cardiomyocytes but not in hepatocytes (Figure 1F)..." However, imaging data from only one cell cannot support the cell-type specific response. One should quantify the difference with multiple cells. Cardiomyocytes were not used for further studies, and I wonder whether the chromatin condensation is relevant to the pharmacological side effects that cause cardiomyopathy and congestive heart failure.

3. U2OS was the main cell line studied in this manuscript. But why were MEFs used in FRAP instead of U2OS?

4. Fig 2C: Adriamycin showed a ring-like structure at 0.5 h time point in HCT116 cell, however, DNA (DAPI) already formed puncta. This is inconsistent with the conclusion that "the condensation of adriamycin could happen first and drive the formation of DNA condensates (line 189).

5. Line 199: the subtitle is ambiguous, not summarizing the results presented in fig S5 accurately.

6. To rule out the possibility that Adriamycin puncta locate in euchromatin, the overlap between Adriamycin and euchromatin markers (H3K43 and H3K27ac) should be examined.

7. The authors drew the conclusion that "...adriamycin-treatment profoundly influence the organization of euchromatin...", but in Figure 3 the condensates prefer to enrich with heterochromatin markers. What's the potential mechanism for euchromatin influenced by adriamycin? If Adriamycin doesn't associate with euchromatin, how come Adriamycin lead to strong chromatin accessibility changes at promoters?

8. Gene expression changes are generally associated with chromatin accessibility. Why etoposide treatment led to massive transcriptional changes but not chromatin accessibility (Fig 4A)?

9. It is interesting that 1,029 TADs lost after adriamycin-treatment (Fig 5D-E). Authors mention that "adriamycin-treatment significantly weakened TAD boundaries throughout the genome, which could profoundly influence gene expression". What gene expression changes are associated with these TAD losses?

10. The Adria-specific TADs showed similar insulation scores between control and adria (Fig 5D, right); these "adria-specific" TADs could be artifacts from data normalization and/or others. They look like common TADs.

11. Fig 5I: the difference between adria and control should be also shown, and compartmental scores for AA and BB should be calculated.

12. Compartmental switches occurred on 18.3% of the genome (Fig 5H). What genes are associated with these compartment changes? Why the authors only showed top 50 genes (Fig 5J) instead of genome-wide pattern?

13. For all the FRAP data, $t_{1/2}$ should be presented.

Reviewer #3 (Remarks to the Author):

The paper by Shi et. al titled "Chemical-Induced Phase Transition and Global Conformational Reorganization of Chromatin" interrogates the cellular response of the chemotherapeutic drug, Adriamycin. The authors use super-resolution microscopy to characterize Adriamycin-induced transition in chromatin organization - from small longitudinal domains into larger and bulkier morphology. These are interpreted as chromatin condensates of mostly heterochromatin, in which Adriamycin is locally enriched at. This transition is shown to affect genome accessibility, transcriptome, and TAD boundaries, have a high affinity for histone H1, and promote chromatin aggregation in-vitro. The authors propose a model whereby Adriamycin binds to histone H1 and promotes phase separation of nucleosomes.

Overall the manuscript employs a multitude of techniques, which altogether provide a comprehensive understanding of Adriamycin-induced effects on chromatin structure and function. The authors convincingly demonstrate that the observed chromatin reorganization is not associated with other known Adriamycin effects related to topoisomerase regulation, DNA damage, and reactive oxygen species. The manuscript contains a considerable amount of effort to provide a clear overview of Adriamycin-induced reorganization, which is admirable. However, there are some significant concerns that need to be addressed, as listed below. These do not necessarily involve major experimental requirements but may require substantial storytelling changes.

(1) Originality - Some of the paper's main findings have been described in the past, yet these are presented as novel findings. This includes past observations on the impact of Adriamycin

(Doxorubicin) on chromatin structure and its interaction with histone H1 (e.g., Bosire, R., Fadel, L., Mocsár, G. et al. Doxorubicin impacts chromatin binding of HMGB1, Histone H1 and retinoic acid receptor. *Sci Rep* 12, 8087 (2022). <https://doi.org/10.1038/s41598-022-11994-z>).

(2) LLPS relevance - While the paper provides valuable complementary data to previous studies, the unifying mechanism is unclear. Firstly, the paper offers minimal support for the involvement of LLPS, which is nevertheless emphasized throughout the paper. Other than the phenotypic morphological transition, which is highly qualitative and shows no liquid phase characteristics (e.g., fusion, coarsening), there is no objective evidence of LLPS. The observation that Adriamycin is being exchanged (FRAP experiments, Figure 2B, line ~180) does not indicate LLPS, but rather that the binding of Adriamycin to chromatin is reversible (as is shown later on in figure 2D and 7C). Moreover, even if LLPS is indeed involved, there's no clear line connecting H1-Adriamycin complexation and the driving forces of phase separation (multivalent interactions, oligomerization etc).

Additional points:

- The authors show that adriamycin response is cell-type specific, yet H1 is not cell-type specific.

Do the authors have any insight on that?

- The line "LLPS of biomolecules, mostly nucleic acids and/or proteins" is inaccurate as there are no native nucleic acids only condensates to the best of my knowledge other than in disease.

- The authors should point out that the super-resolution microscopy data is with fixed cells.

Showing that the same structures are observed in live cells is important.

- There is no correlation between STED and EM images – The first show structural reorganization throughout the nucleus, while the latter shows it only affects the nuclear lamina.

Point-to-point response

We would like to thank three reviewers for their constructive and expert comments on improving our manuscript. In the revised manuscript, we have performed additional experiments to address the excellent questions raised by reviewers. To respond reviewers' points, we also rephrased some statement/hypothesis of the manuscript for the preciseness and clarity. We hope the reviewers will be convinced that all his/her concerns have been addressed. The figures cited and the line numbers in the response below are according to the numeration of the figures and line numbers in the revised manuscript unless particularly mentioned.

Reviewer #1:

The study applied a small library of molecules targeting genomic DNA to examine the physical state of (viscoelastic) chromatin by STED and electron microscopy. They found that Adriamycin induced spatially resolved DNA puncta – preferentially affecting euchromatin, and independently of Adriamycin effects on ROS, ferroptosis, the TOP2-DNA ligase, or cell death. Puncta were induced in a cell type specific way (cardiac myocytes but not hepatocytes), and they were reversible after removal of agonist - suggesting physiologic relevance. In affected cells, adriamycin was found to colocalize exactly with the DNA puncta implicating direct interaction with components of the puncta – and this enabled studies showing the Adriamycin-induced puncta had features of liquid-like biomolecular condensates - unlike those reported before. Notably, the formation of adriamycin-condensates was closely correlated with structural changes in organization and accessibility of chromatin – and gene expression.

The evidence for effects on chromatin structure and function appears to be strong – though it should be reviewed by someone working in the field, which I am not. I do find the evidence for condensate definitive – features of liquid-like behavior are convincingly demonstrated; and the condensates are reproduced in vitro with known components. The condensates are shown to result mechanistically by direct interaction with chromatin (not DNA) – by interaction with H1 histone specifically. This, and the close correlation between condensate formation and chromatin structure and function, are meaningful contributions to the field.

Overall, the studies are well conceived, technically well done and well controlled, and they are reasonably interpreted. Though why condensate formation may be important to regulation of chromatin structure is not made clear. These ideas might be discussed. We thank the reviewer for the positive evaluation of our study. As indicated in the Introduction, our present study aims at investigation how the physical property of chromatin influences the three-dimensional conformation, gene expression, or biological function. In the revised manuscript, we included more discussion on the viscoelastic property of condensates we observed, as well as its impact on the chromatin structure and gene expression, to echo the beginning (Line #496-500). We thank the

reviewer for this insightful suggestion.

Line 112 – use of word “superb” is commentary – perhaps substitute “high-resolution”
We thank the reviewer for the suggestion and have changed it accordingly.

Line 151 – use of term “condensates” here (and following) not yet justified – evidence for condensate formation is not yet presented until Fig 2 and supplemental data.
We thank the reviewer for this suggestion. We have replaced “condensates” in those paragraphs with “structures” before showing the formation results. We also defined the meaning of “condensates” in the text (Line #183-184) for the preciseness.

Line 248-249 – and 258-260 – sentences need rewording – hard to understand.
We’re sorry for those distractive descriptions and have reworded those sentences. For the former sentence, we moved it forward (Line #254-256) to describe the possible limitation of this live imaging approach. For the latter sentence, we removed this DNA damage discussion since it will be discussed later in the text (Line #293-295).

Line 397 – figure reference is confusing.
We’re sorry for the mistake and have changed it to 7E.

Reviewer #2:

The manuscript by Shi et al. reports that adriamycin induces the condensation of chromatin and influences the 3D genome organization. They screened out adriamycin from different types of DNA targeting chemicals as it induces chromatin condensation which is dynamic and reversible, co-localized with heterochromatin markers. Interestingly, this change is independent of its TOP2 poisoning activity, DNA damage, ROS, and programmed cell death. In addition, after Adriamycin treatment, the loss of TAD boundaries, decrease of chromatin accessibilities and changes of gene expression were revealed by genomic data such as Hi-C and ATAC-seq. The authors further indicated that adriamycin interacts with histone H1 and induces phase transition. Overall, this is an interesting and comprehensive study and the manuscript was well-written. Below are some suggestions which could help the authors to improve their manuscript.

We thank the reviewer for those constructive suggestions and have made corresponding changes to improve our manuscript.

1. The mechanism on how Adriamycin regulate chromatin structure remain unclear. The working model (Fig 7G) isn't well supported by Fig 7, which presents mainly in-vitro data. How about the interaction of Adriamycin and histone H1 in live cells? The adding of Adriamycin lead to the formation of H1 condensates in cells? Are those potential H1 condensates associated with the puncta as observed in Fig 1?

We thank the reviewer for this constructive suggestion. To trace the interaction between

adriamycin and H1 *in vivo*, we performed immunostaining on histone H1 in adriamycin-treated U2OS cells at short time intervals (Fig. 7h). Consistent with the *in vitro* finding, we found H1 formed puncta that strongly co-localized with adriamycin and DNA (Fig. 7h). The condensation of adriamycin, H1, and DNA happened almost simultaneously, as demonstrated by the similar distribution patterns at all time points. These results suggest that adriamycin regulates chromatin structure by driving the formation of adriamycin-H1-DNA condensates.

Fig. 7h. The co-localization of adriamycin and histone H1 in cells. U2OS cells were treated with 1.5 $\mu\text{g/ml}$ adriamycin, fixed at indicated time points, and stained with anti-H1 antibody and DAPI. Scale bar, 10 μm .

2. The authors stated that “...adriamycin induced drastic chromatin condensation in cardiomyocytes but not in hepatocytes (Figure 1F)...” However, imaging data from only one cell cannot support the cell-type specific response. One should quantify the difference with multiple cells. Cardiomyocytes were not used for further studies, and I wonder whether the chromatin condensation is relevant to the pharmacological side effects that cause cardiomyopathy and congestive heart failure.

We thank the reviewer for raising this important issue. To solidify the conclusion of cell-type specificity, we quantified the ratio of cardiomyocytes or hepatocytes exhibited condensed chromatin in four cell populations, including cardiomyocytes, non-cardiomyocytes, hepatocytes, and non-hepatocytes (Fig. 1h, also see later for the discussion). Interestingly, we observed shape differences in the intracellular level of adriamycin between cardiomyocytes and hepatocytes: the level of adriamycin in cardiomyocytes (Tnni3^+) are similar to adjacent non-cardiomyocytes (Tnni3^-), while the level of adriamycin in hepatocytes (Alb^+) are much lower than non-hepatocytes (Alb^-) in the same fields (Fig. 1h). The low level of adriamycin accumulated in hepatocytes, which could be attributed to the drug effluxion ability of hepatocytes^{1,2}, is consistent with the minimal chromatin condensation in hepatocytes (Fig. 1g). To further investigate the correlation between the chromatin condensation and cardiotoxicity, we treated primary cardiomyocytes with two clinical-used anthracyclines with high and low cardiotoxicity, daunorubicin and aclarubicin,

respectively (Supplementary Fig. 3d)^{3,4}. Daunorubicin, as adriamycin, induced strong chromatin condensation in primary cardiomyocytes, while aclarubicin barely induced chromatin condensation (Supplementary Fig. 3d). Notably, although this result suggests a link between the chromatin condensation and cardiotoxicity, we couldn't trace the intracellular level of aclarubicin due to its low fluorescence under the microscope (data not shown). Thus, the causative link between the chromatin condensation effect and the cardiotoxicity may only be established with more structural insights on the adriamycin-chromatin interaction (such as ablation of their interaction by changing function groups of adriamycin), which could be out of the scope of the current study.

Fig. 1g and 1h. Adriamycin induced significant chromatin condensation in primary cardiomyocytes. (g) Primary cells isolated from P1 mice were cultured for 3-5 days, followed by 1.5 $\mu\text{g/ml}$ adriamycin treatment for 4 h. Cells were immunostained with cardiomyocyte (Tnni3) and hepatocyte (Alb) markers, as well as DAPI. Cells positive for adriamycin were examined for their chromatin conformation. (h) The differential accumulation of adriamycin in different cell populations. Dashed circles indicate the nuclear outlines of cardiomyocytes (Tnni3⁺) or hepatocytes (Alb⁺). Yellow arrows indicate non-cardiomyocytes (Tnni3⁻) or non-hepatocytes (Alb⁻). Scale bar, 10 μm . The ratios of cells showing condensed chromatin (as in 1G) in those four populations are shown above each bar. ns, not significant; ***, $P < 0.001$.

Supplementary Fig. 3d. Chromatin structures of daunorubicin- and aclarubicin-treated primary cardiomyocytes. Primary cardiomyocytes were treated by 2 μ M daunorubicin and 5 μ M aclarubicin for 4 h. Cardiomyocytes positive for Tnni3 were examined for their chromatin conformation. Scale bar, 10 μ m. ****, $P < 0.0001$.

3. U2OS was the main cell line studied in this manuscript. But why were MEFs used in FRAP instead of U2OS?

We thank the reviewer for raising this question. The adriamycin itself possesses a light-activated cytotoxicity (phototoxicity), possibly through the enhanced production of reactive oxygen species⁵. In our hands, we found adriamycin-treated MEFs are more resistant to microscopic lasers used in live imaging compared with adriamycin-treated U2OS cells. We, therefore, employed MEFs for FRAP experiments. During the revision, we also attempted to perform FRAP on U2OS by adjusting imaging parameters (Fig. R1). Compared with MEFs, U2OS cells exhibited smaller, more homogenous adriamycin condensates, on which we also observed the fluorescence recovery with faster dynamics, suggesting the fluidity of adriamycin within condensates in both cell types. Since we couldn't justify the influence of phototoxicity on adriamycin dynamics in U2OS cells, we appended the result here for the reviewer's reference.

Fig. R1. FRAP experiment showing the exchange dynamics of adriamycin in condensates. U2OS cells were treated with 1.5 mg/ml adriamycin for 4 h before subjecting to FRAP analysis. $t_{1/2}$ are indicated above the intensity graphs. Scale bar, 10 μ m.

4. Fig 2C: Adriamycin showed a ring-like structure at 0.5 h time point in HCT116 cell, however, DNA (DAPI) already formed puncta. This is inconsistent with the conclusion that “the condensation of adriamycin could happen first and drive the formation of DNA condensates (line 189).

We thank the reviewer for his/her careful reading. We agree that there seems to be cell-type specificity in the dynamics of adriamycin-DNA condensate formation. We have removed the sentence to avoid the overstatement.

5. Line 199: the subtitle is ambiguous, not summarizing the results presented in fig S5 accurately.

We have rephrased the subtitle to make it more specific.

6. To rule out the possibility that Adriamycin puncta locate in euchromatin, the overlap between Adriamycin and euchromatin markers (H3Kme3 and H3K27ac) should be examined.

We thank the reviewer for this insightful suggestion and performed immunostaining on adriamycin-treated U2OS cells with anti-H3K4me3 and anti-H3K27ac antibodies (Fig. 3e and 3f). Interestingly, adriamycin condensates showed a mostly non-overlapped pattern with H3K4me3 signals (promoters, Fig. 3e) but were overlapped with H3K27ac signals (promoters and enhancers, Fig. 3f). Those results suggest that besides of the major association with the heterochromatin, adriamycin could also associate with the enhancer region of the euchromatin. This observation also explains the influence of adriamycin on genes located in the euchromatin (see the next question).

Fig. 3e and 3f. The association of adriamycin and euchromatin regions. Adriamycin-chromatin condensates were localized mutual exclusively with H3K4me3 (promoters), but partially co-localized with euchromatin marker H3K27ac (promoters and enhancers). Arrows indicate adriamycin condensates. Scale bar, 10 μ m.

7. The authors drew the conclusion that “...adriamycin-treatment profoundly influence the organization of euchromatin...”, but in Figure 3 the condensates prefer to enrich with heterochromatin markers. What’s the potential mechanism for euchromatin influenced by adriamycin? If Adriamycin doesn’t associate with euchromatin, how come Adriamycin lead to strong chromatin accessibility changes at promoters?

We thank the reviewer for raising this point. The H3K27ac staining (Fig. 3f) argues that adriamycin could also influence the organization of euchromatin by, at least in part, associating with enhancer regions. Additionally, the association of adriamycin with euchromatin could be underestimated due to its loose conformation. We have modified our conclusion, re-drew the model, and included the discussion in the revised manuscript.

8. Gene expression changes are generally associated with chromatin accessibility. Why etoposide treatment led to massive transcriptional changes but not chromatin accessibility (Fig 4A)?

We thank the reviewer for pointing out this problem. We would first like to clarify that etoposide treatment resulted in ~1300 genes exhibiting significant expression changes, significantly fewer than the number of differentially expressed genes upon adriamycin treatment (~2500). Therefore, the phrase “massive transcriptional changes” we used to describe the effects of etoposide treatment may not be appropriate, which we have now removed from the revised manuscript. In this revision, we carefully repeated the ATAC-seq experiment on control- and etoposide-treated U2OS cells (Fig. 4a) to examine the chromatin accessibility. With an improved consistency between technical repeats, we found 150 gained and 72 lost peaks in etoposide-treated cells, more than the previous result but still much less than the gained/lost peaks induced by adriamycin (9,456/11,747). The more significant alterations in chromatin accessibility induced by adriamycin are in line with the observations that only adriamycin, but not etoposide treatments caused the formation of chromatin condensates. Thus, adriamycin affects both local chromatin accessibility and higher-order chromatin organization to a greater extent than etoposide.

As the reviewer pointed out, the chromatin accessibility is closely associated with transcription initiation ⁶. Yet, the level of a matured mRNA is reflected by many other factors, such as the pause-release of the RNA polymerase, termination, RNA processing, RNA degradation, even the 3D chromatin conformation ⁷. It has been demonstrated that Top2 (Top2a and/or Top2 β), which is targeted and degraded by etoposide, could modulate the pause and release of Pol II ^{8,9}, the local chromatin architecture for transcription activation ¹⁰, and the genome organization through associating with cohesin and CTCF ¹¹. Thus, the transcriptional changes induced by etoposide could be the consequence of both chromatin accessibility alteration and Top2 inhibition.

9. It is interesting that 1,029 TADs lost after adriamycin-treatment (Fig 5D-E). Authors mention that “adriamycin-treatment significantly weakened TAD boundaries throughout the genome, which could profoundly influence gene expression”. What gene expression changes are associated with these TAD losses?

We thank the reviewer for raising this important point. We did not specifically look into the genes associated with the lost TAD boundaries for the following reasons: (1) The loss or weakening of a TAD boundary may influence the expression of genes located at a distance from the boundary, sometimes several hundred kilobases away, by altering the distribution of chromatin contacts; (2) In addition to the lost TAD boundaries, the shared TAD boundaries also exhibit a significant increase in the insulation score, indicating a decrease in the insulation ability (Supplementary Fig. 9a). Thus, adriamycin treatment resulted in a global weakening of TAD boundaries that affect most TADs throughout the genome.

To assess whether the gene expression changes upon adriamycin treatment are related to the weakening of TAD boundaries, we selected ~300 lost TAD boundaries that exhibited the greatest increase in the insulation score (Group1, insulation changes > 0.2) and ~300 shared TAD boundaries that exhibited the least increase in the insulation score (Group2, insulation changes < 0.01), and examined the expression changes for genes located within 200 kb or 100 kb from these two groups of TAD boundaries. We found that the differentially expressed genes are more enriched within the vicinity of the Group1 TAD boundaries than the Group2 ones (Supplementary Fig. 9b and 9c). Thus, the changes in TAD organization may partially contribute to the gene expression changes by altering chromatin interaction landscapes.

Supplementary Fig. 9. Changes in TAD organization contribute to gene expression shifts. (a) The boxplot summarizes the changes in insulation scores at the control-specific, shared, and adria(adriamycin)-specific TAD boundaries shown in Figure 5E. Increases in insulation scores indicate the weakening of the TAD boundaries. The median of insulation changes for each class of TAD boundaries is indicated below the plot. Note that both the control-specific and the shared TAD boundaries are substantially weakened upon adriamycin treatment. Centerline, median; box limits, upper and lower quartiles; whiskers, 1.5× interquartile range. (b) To assess the relationship between TAD boundary changes and gene expression changes, 375 control-specific TAD boundaries that exhibit an insulation change greater than 0.2 and 322 shared TAD boundaries that exhibit an insulation change less than 0.01

were selected and names as “Group1” and “Group2”. Centerline, median; box limits, upper and lower quartiles; whiskers, $1.5\times$ interquartile range. (c) Pie charts illustrates the quantity of differentially expressed genes and stable genes found within each class of TAD boundary with the upstream and downstream expansion of 200 kb (up) and 100 kb (down). Chi-square tests were conducted to determine the enrichment of differentially expressed genes compared to stably expressed genes within TAD boundaries.

10. The Adria-specific TADs showed similar insulation scores between control and adria (Fig 5D, right); these “adria-specific” TADs could be artifacts from data normalization and/or others. They look like common TADs.

As the reviewer pointed out, the “Adria-specific” TAD boundaries represent a set of TAD boundaries that are marginally changed upon adriamycin treatment (medium insulation changes = -0.03, Supplementary Fig. 9), but are only identified as TAD boundaries in the treated samples. Such a difference in TAD boundary annotation may be attributed to several reasons: (1) At some shallow TAD boundaries, a slight reduction of boundary strength may render them below the detection threshold; (2) Some TAD boundaries exhibit a “flatter” valley bottom, and the local insulation minima at the bottom may vary in different Hi-C datasets, leading to differences in the assignment of TAD boundary positions. Either way, these “Adria-specific” TAD boundaries are likely resulted from the variations in Hi-C data, rather than the data processing artifacts.

It is certainly possible that these “Adria-specific” TAD boundaries originate from experimental variations in Hi-C datasets, given that their insulation changes are marginal. However, it is also challenging to definitively exclude the possibility that these TAD changes reflect subtle biological differences in chromatin structure. For the sake of convenience, we have retained the term “Adria-specific” TADs in our manuscript. As we stated in the manuscript, the main conclusion of this section is that “adriamycin-treatment causes global weakening of TAD boundaries throughout the genome”, which is supported by the substantial changes in insulation scores for both the “control-specific” and “shared” TAD boundaries. Therefore, we believe the retention of “Adria-specific” TAD boundaries does not affect our conclusion.

11. Fig 5I: the difference between adria and control should be also shown, and compartmental scores for AA and BB should be calculated.

We thank the reviewer for the suggestion. We have added the following compartment scores to Fig. 5i to indicate the strength of interactions between genomic regions belonging to the same compartment or different compartments:

Control: AA: 1.198, BB: 1.119, AB: 0.820

Adria: AA: 1.113, BB: 1.163, AB: 0.844.

We have also incorporated a new panel to illustrate the ratio of saddle plot data between

Adriamycin-treated and control samples (Fig. 5i, right). It can be readily appreciated that while the AA interactions decrease, the BB and AB interactions increase upon adriamycin treatment, consistent with the strengthening of the B compartment.

Fig. 5i. The change of contact frequencies between compartments upon adriamycin treatment. Genomic regions belong to the B compartment (B-B) exhibited a notable increase.

12. Compartmental switches occurred on 18.3% of the genome (Fig 5H). What genes are associated with these compartment changes? Why the authors only showed top 50 genes (Fig 5J) instead of genome-wide pattern?

We thank the reviewer for raising this question. In the revised manuscript, we plotted all the differentially expressed genes ($\text{abs}(\log_2\text{Foldchange}) > 2$, $\text{P}_{\text{adj}} < 0.05$) against their Eigen1 (E1) values to indicate compartment switches. Consistent with the result of top 50 genes, we noticed a significant ($p < 0.22e-16$) correlation between A-B compartment switch (indicated by the decreased E1) and gene down-regulation, but not gene up-regulation ($p = 0.7608$) (Fig. 5j). For those genes, gene ontology analysis revealed the enrichment of apoptosis, NF-kB, and TNF pathways (Fig. 5j).

Fig. 5j. Up, the boxplots showing $\log_2(\text{fold change})$ Eigen1 (E1) values of all the differentially expressed genes with $\text{abs}(\log_2\text{Foldchange}) > 2$ and $\text{P}_{\text{adj}} < 0.05$. Adriamycin treatment didn't induce statistically significant

compartment changes for up-regulated genes, but led to the significant decrease of E1 values for down-regulated genes. The whole gene bodies as well as 2000 bp upstream of TSS sites were used to overlap with E1 values of 10 kb binned genomic regions. Gene level E1 value were then calculated using averaged E1 of corresponding gene regions. Down, KEGG analysis of the pathways enriched in down-regulated genes with decreased E1.

13. For all the FRAP data, $t_{1/2}$ should be presented.

We have added $t_{1/2}$ to FRAP results.

Reviewer #3:

The paper by Shi et. al titled "Chemical-Induced Phase Transition and Global Conformational Reorganization of Chromatin" interrogates the cellular response of the chemotherapeutic drug, Adriamycin. The authors use super-resolution microscopy to characterize Adriamycin-induced transition in chromatin organization - from small longitudinal domains into larger and bulkier morphology. These are interpreted as chromatin condensates of mostly heterochromatin, in which Adriamycin is locally enriched at. This transition is shown to affect genome accessibility, transcriptome, and TAD boundaries, have a high affinity for histone H1, and promote chromatin aggregation in-vitro. The authors propose a model whereby Adriamycin binds to histone H1 and promotes phase separation of nucleosomes.

Overall the manuscript employs a multitude of techniques, which altogether provide a comprehensive understanding of Adriamycin-induced effects on chromatin structure and function. The authors convincingly demonstrate that the observed chromatin reorganization is not associated with other known Adriamycin effects related to topoisomerase regulation, DNA damage, and reactive oxygen species. The manuscript contains a considerable amount of effort to provide a clear overview of Adriamycin-induced reorganization, which is admirable. However, there are some significant concerns that need to be addressed, as listed below. These do not necessarily involve major experimental requirements but may require substantial storytelling changes.

We're grateful for the reviewer's positive evaluation. We have revised the manuscript to address those concerns and advices.

(1) Originality – Some of the paper's main findings have been described in the past, yet these are presented as novel findings. This includes past observations on the impact of Adriamycin (Doxorubicin) on chromatin structure and its interaction with histone H1 (e.g., Bosire, R., Fadel, L., Mocsár, G. et al. Doxorubicin impacts chromatin binding of HMGB1, Histone H1 and retinoic acid receptor. *Sci Rep* 12, 8087 (2022). <https://doi.org/10.1038/s41598-022-11994-z>).

We thank the reviewer for raising this important issue. These two observations – (1) the

impact of adriamycin on chromatin structure and (2) adriamycin interacts with histone H1 – are actually (seemly) contradictory in past studies. In the original manuscript, we cited the key findings on these two observations to discuss about the influence of adriamycin on the chromatin structure (Pang *et al.*, *Nat Comm*, 2013), as well as the interaction between adriamycin and H1 (Zarger *et al.*, *Int J Biol Macromol*, 2000). The study by Bosire *et al.* is in line with former study by Pang *et al.*, arguing that adriamycin interferes with the interaction between histone and DNA, therefore displacing histones from chromatin (histone eviction). Our present study demonstrated a strong interaction between adriamycin and histone H1, as well as the formation of adriamycin-H1-DNA complexes *in vitro* and *in vivo*, more consistent with the latter study by Zarger *et al.*, although we can also observe the chromatin eviction effect in certain circumstances. In the first submission, we attempted to reconcile these two observations (Line #450-463 of the original manuscript), suggesting that “one possible explanation is that adriamycin could initially displace or evict H1 from chromatin, and the displaced or free H1 could form the condensates with adriamycin and re-associate with chromatin, leading to the re-organization of its 3D conformation. Alternatively, adriamycin could directly associate with DNA-bound H1 in a site-specific manner.” Thus, although these two observations are not new, the major contribution of our study is (1) confirming that formation of adriamycin-H1-DNA condensates *in vitro* and *in vivo*, and (2) systematically characterizing the impact of adriamycin-H1-DNA interaction on high-order chromatin structures and chromatin accessibilities (please also see below for the LLPS discussion). In the revised manuscript, we have included the study by Bosire *et al.* and modified the paragraph of discussion to make the point as clear as possible.

(2) LLPS relevance - While the paper provides valuable complementary data to previous studies, the unifying mechanism is unclear. Firstly, the paper offers minimal support for the involvement of LLPS, which is nevertheless emphasized throughout the paper. Other than the phenotypic morphological transition, which is highly qualitative and shows no liquid phase characteristics (e.g., fusion, coarsening), there is no objective evidence of LLPS. The observation that Adriamycin is being exchanged (FRAP experiments, Figure 2B, line ~180) does not indicate LLPS, but rather that the binding of Adriamycin to chromatin is reversible (as is shown later on in figure 2D and 7C). Moreover, even if LLPS is indeed involved, there's no clear line connecting H1-Adriamycin complexation and the driving forces of phase separation (multivalent interactions, oligomerization etc).

We thank the reviewer for raising this crucial point. We agree that the condensates formed by adriamycin and chromatin are not typical LLPS considering its special morphology and property. In the original manuscript, we intentionally avoided using LLPS to describe the condensates or aggregates formed by adriamycin and the chromatin in cells. In the Introduction of the first submission, we discussed about debates on the property of heterochromatin regarding whether its liquid-like, solid-like, or “viscoelastic-like”. As mentioned in the original Introduction, the aim of this study is to investigate “how the physical property of chromatin influences the three-dimensional conformation, gene expression, or biological effect”. In agreement of the

reviewer, we also described the FRAP results as the indication of that “adriamycin within condensates could diffuse and exchange with other adriamycin molecules in the nucleoplasm” (Fig. 2d). For the *in vitro* experiments, we used “phase transition” as the term to describe the formation of fibrous aggregates or condensates without indicating any specific phase (Fig. 7c). Actually, we did observe an “expansion-mediated fusion” of some fibrous aggregates *in vitro* (Fig. 7f), distinctive for the typical “fusion” phenomenon observed in other LLPS molecules. We think the spectacular morphology and slow dynamics of those aggregates reflect a more viscoelastic-like state, although more studies are required for identifying the physical property of those condensates. In the revised manuscripts, we adjusted the descriptions and added more discussion to emphasize this difference.

Fig. 7e-f. The dynamics of condensation of adriamycin and H1-CFP. (e) By itself, neither adriamycin nor H1-CFP formed condensate. Mixing adriamycin and H1-CFP led to the phase separation, forming condensates containing both adriamycin and H1-CFP. Scale bar, 10 μm . (f) The “expansion-mediated fusion” dynamics of adriamycin condensates. Scale bar, 1 μm . Arrows indicate “seeds” that expanded with time.

We also agree that the driving force of phase separation adriamycin-H1 is not clear. To further investigate this issue, we predicted the structure of histone H1 by PSIPRED⁵. In the structural prediction, the N-terminus and C-terminus of histone H1 are disordered regions (Supplementary Fig. 10a). We found that the disordered C-terminal fragment of H1 (H1-3) has much higher affinity than the N-terminal (H1-1) and the ordered helix (H1-2) regions (Supplementary Fig. 10b), and was able to form condensates with adriamycin *in vitro* (Supplementary Fig. 10c). Those results suggest that the interaction

between the C-terminal disordered region of H1 and adriamycin could be the driving force for the phase transition. We have included this result in the revised manuscript.

Supplementary Fig. 10. Adriamycin interacts with the C-terminal region of H1 for phase separation. (a) The structure of H1 was predicted by PSIPRED and dissected to three fragments with 27, 56, and 111 amino acids. Histone H1 fragments fusion with CFP. The fusion proteins were expressed in *E. Coli* and purified by Ni columns. (b) SPR analysis revealed the affinities between adriamycin and H1 fragments, in which the C-terminal of H1 (H1-3) exhibited the highest affinity. (c) *In vitro* condensation of adriamycin and H1 fragments. Mixing adriamycin and H1-3 fusion protein led to the phase separation, forming condensates containing both adriamycin and H1-CFP. Scale bar, 10 μ m.

Additional points:

- The authors show that adriamycin response is cell-type specific, yet H1 is not cell-type specific. Do the authors have any insight on that?

We thank the reviewer for raising this interesting question. In the revised manuscript, we included new results showing sharp differences in the intracellular level of adriamycin between cardiomyocytes and hepatocytes (Fig. 1h): the level of adriamycin in cardiomyocytes (Tnni3⁺) are similar to adjacent non-cardiomyocytes (Tnni3⁻), while

the level of adriamycin in hepatocytes (Alb⁺) are much lower than adjacent non-hepatocytes (Alb⁻) (Fig. 1h). The low level of adriamycin accumulated in hepatocytes, which could be due to the efflux of the drug, is consistent with the minor chromatin condensation in hepatocytes (Fig. 1g and 1h), although we couldn't exclude other possible mechanism(s) for cell-type specificities which merits further investigations.

Fig. 1g and 1h. Adriamycin induced significant chromatin condensation in primary cardiomyocytes. (g) Primary cells isolated from P1 mice were cultured for 3-5 days, followed by 1.5 $\mu\text{g/ml}$ adriamycin treatment for 4 h. Cells were immunostained with cardiomyocyte (Tnni3) and hepatocyte (Alb) markers, as well as DAPI. Cells positive for adriamycin were examined for their chromatin conformation. (h) The differential accumulation of adriamycin in different cell populations. Dashed circles indicate the nuclear outlines of cardiomyocytes (Tnni3⁺) or hepatocytes (Alb⁺). Yellow arrows indicate non-cardiomyocytes (Tnni3⁻) or non-hepatocytes (Alb⁻). Scale bar, 10 μm . The ratios of cells showing condensed chromatin (as in 1G) in those four populations are shown above each bar. ns, not significant; ***, $P < 0.001$.

- The line "LLPS of biomolecules, mostly nucleic acids and/or proteins" is inaccurate as there are no native nucleic acids only condensates to the best of my knowledge other than in disease.

We thank the reviewer for pointing out this ambiguity issue, which we actually mean RNAs (such as rRNAs or some noncoding RNAs). We indicated it clearly in the revised manuscript.

- The authors should point out that the super-resolution microscopy data is with fixed

cells. Showing that the same structures are observed in live cells is important. We thank the reviewer for this critical question. In the revised manuscript, we made efforts to image live cells by the structured illumination microscopy (SIM) which, consistent with STED results, also revealed clear adriamycin-chromatin condensates (Figure 2D).

Fig. 2d. The adriamycin-DNA condensates in live cells revealed by SIM microscopy. U2OS cells were pre-treated with 5 μ M SiR-DNA for overnight, followed by the treatment of 1.5 μ g/ml adriamycin (adria) for 2 hours before imaged on Zeiss Elyra7 SIM super-resolution microscope. Scale bar, 10 μ m.

- There is no correlation between STED and EM images – The first show structural reorganization throughout the nucleus, while the latter shows it only affects the nuclear lamina.

We thank the reviewer for the careful reading. We performed new IF experiments to compare with the EM results. In EM images, we actually observed condensates in the nucleoplasm (Figure 1E), consistent with the LM imaging for Lamin A/C and DNA (Figure 1F). As indicated in the figure, the chromatin condensates were located in both the nucleoplasm and the proximity of nuclear lamina (Figure 1E and 1F). Overall, there were indeed fewer condensates in EM images compared with LM ones, which could be due to the extra-thin sections employed by EM. We have zoomed nucleoplasmic condensates in EM images, included the immunofluorescence staining results, and discussed those differences in the revised manuscript.

Fig. 1e and 1f. Chromatin condensates induced by adriamycin are located in both the nucleoplasm and the nuclear peripheral. (e)

Electron microscopy revealed different morphologies of chromatin condensates. Scale bar, 10 μ m. White arrows indicate the condensates formed in the proximity of nuclear envelopes. (f) Immunostaining of Lamin A/C (red) with DAPI staining (green) in adriamycin-treated U2OS cells. Scale bar, 10 μ m. White arrows indicate the condensates formed in the proximity of nuclear envelopes.

Reference

1. Colabufo, N.A., Berardi, F., Contino, M., Niso, M. & Perrone, R. ABC pumps and their role in active drug transport. *Curr Top Med Chem* **9**, 119-29 (2009).
2. Honjo, Y. *et al.* Acquired mutations in the MXR/BCRP/ABCP gene alter substrate specificity in MXR/BCRP/ABCP-overexpressing cells. *Cancer Res* **61**, 6635-9 (2001).
3. Haruyama, H. *et al.* [Cardiotoxicity of daunorubicin and aclacinomycin A in patients with acute leukemia]. *Gan To Kagaku Ryoho* **9**, 516-21 (1982).
4. Qiao, X. *et al.* Uncoupling DNA damage from chromatin damage to detoxify doxorubicin. *Proc Natl Acad Sci U S A* **117**, 15182-15192 (2020).
5. Greco, G. *et al.* Light-Enhanced Cytotoxicity of Doxorubicin by Photoactivation. *Cells* **12**(2023).
6. Zaret, K.S. & Carroll, J.S. Pioneer transcription factors: establishing competence for gene expression. *Genes Dev* **25**, 2227-41 (2011).
7. Kim, S. & Shendure, J. Mechanisms of Interplay between Transcription Factors and the 3D Genome. *Mol Cell* **76**, 306-319 (2019).
8. Bunch, H. *et al.* Transcriptional elongation requires DNA break-induced signalling. *Nat Commun* **6**, 10191 (2015).
9. Herrero-Ruiz, A. *et al.* Topoisomerase II α represses transcription by enforcing promoter-proximal pausing. *Cell Rep* **35**, 108977 (2021).
10. Ju, B.G. *et al.* A topoisomerase II β -mediated dsDNA break required for regulated transcription. *Science* **312**, 1798-802 (2006).
11. Canela, A. *et al.* Genome Organization Drives Chromosome Fragility. *Cell* **170**, 507-521.e18 (2017).

REVIEWERS' COMMENTS:

Reviewer #2 (Remarks to the Author):

My concerns have been well addressed. I think this revised manuscript is now suitable for publication in Nature Communications.

Reviewer #3 (Remarks to the Author):

Overall, the revised manuscript is notably improved, and the authors have clearly put significant thought and effort into addressing reviewer comments. However, there are a few persisting issues:

In regards to my previous point #1 (I.e., originality) - the text has been improved and clarified. However, the manuscript is still written as if the phenotypical structural transition of DNA caused by Adriamycin is a new finding.

For instance, the abstract says "We found that adriamycin (doxorubicin), a widely-used chemotherapeutic drug against malignancies, is able to induce local condensation and global conformational change of chromatin in cancer and primary cells".

Another example (line 90): "We found an anthracycline antibiotic and chemotherapeutic drug, adriamycin (doxorubicin), that can unexpectedly induce the condensation of chromatin at the microscale in cancer and primary cells in a reversible and cell-type-specific manner."

Namely, neither in the abstract nor the introduction, there is any acknowledgment that Adriamycin has ever been reported to have any effect on DNA. Nevertheless, a clear visual effect has been observed in cells. A clearer distinction between the well-documented phenotypical transition and the new mechanistic observations should be made.

- A smaller point that nevertheless needs another touch has to do with the composition of native condensates (i.e., the line: "LLPS of biomolecules, mostly nucleic acids and/or proteins"). The authors changed the text to "mostly RNAs and/or proteins", yet it would be more accurate to replace the order with "mostly proteins and/or nucleic acids". This is since all native condensates reported to date contain proteins, whereas, nucleic acids (inc RNA) are not found in all of them.

Point-to-point response

We thank two reviewers for their comments. In this revision, we have made corresponding changes to address the point raised by Reviewer #3. We hope the reviewer will be convinced that all his/her concern has been adequately addressed.

Reviewer #2:

My concerns have been well addressed. I think this revised manuscript is now suitable for publication in Nature Communications.

We deeply thank the reviewer for his/her suggestions and positive evaluation on our study.

Reviewer #3:

Overall, the revised manuscript is notably improved, and the authors have clearly put significant thought and effort into addressing reviewer comments. However, there are a few persisting issues:

In regards to my previous point #1 (I.e., originality) - the text has been improved and clarified. However, the manuscript is still written as if the phenotypical structural transition of DNA caused by Adriamycin is a new finding.

We agree and thank the reviewer for this constructive comment and have modified the following descriptions accordingly.

For instance, the abstract says "We found that adriamycin (doxorubicin), a widely-used chemotherapeutic drug against malignancies, is able to induce local condensation and global conformational change of chromatin in cancer and primary cells".

We have changed the sentences as the following:

“This microscopic screening approach reveals that adriamycin (doxorubicin), a widely-used anticancer drug that is known to interfere with the chromatin, specifically induces visible local condensation and global conformational change of chromatin in cancer and primary cells. Hi-C and ATAC-seq experiments systematically and quantitatively

Another example (line 90): "We found an anthracycline antibiotic and chemotherapeutic drug, adriamycin (doxorubicin), that can unexpectedly induce the condensation of chromatin at the microscale in cancer and primary cells in a reversible and cell-type-specific manner."

We have changed these descriptions as the following:

“This phenotypical screening reveals an anthracycline antibiotic and chemotherapeutic drug known to bind and interfere with the chromatin, adriamycin (doxorubicin), induces the condensation of chromatin in live and fixed cells in a visible, reversible, and cell-type-specific manner.”

Namely, neither in the abstract nor the introduction, there is any acknowledgment that Adriamycin has ever been reported to have any effect on DNA. Nevertheless, a clear visual effect has been observed in cells. A clearer distinction between the well-documented phenotypical transition and the new mechanistic observations should be made.

We agree with the reviewer and have added those points in the abstract and the introduction.

- A smaller point that nevertheless needs another touch has to do with the composition of native condensates (i.e., the line: "LLPS of biomolecules, mostly nucleic acids and/or proteins"). The authors changed the text to "mostly RNAs and/or proteins", yet it would be more accurate to replace the order with "mostly proteins and/or nucleic acids". This is since all native condensates reported to date contain proteins, whereas, nucleic acids (inc RNA) are not found in all of them.

We thank the reviewer for the suggestion and have changed it accordingly.